# Policy Improvement via Imitation of Multiple Oracles

**Ching-An Cheng**
Microsoft Research
Redmond, WA 98052
chinganc@microsoft.com

**Andrey Kolobov**
Microsoft Research
Redmond, WA 98052
akolobov@microsoft.com

**Alekh Agarwal**
Microsoft Research
Redmond, WA 98052
alekha@microsoft.com

## Abstract

Despite its promise, reinforcement learning's real-world adoption has been hampered by the need for costly exploration to learn a good policy. Imitation learning (IL) mitigates this shortcoming by using an oracle policy during training as a bootstrap to accelerate the learning process. However, in many practical situations, the learner has access to multiple suboptimal oracles, which may provide conflicting advice in a state. The existing IL literature provides a limited treatment of such scenarios. Whereas in the single-oracle case, the return of the oracle's policy provides an obvious benchmark for the learner to compete against, neither such a benchmark nor principled ways of outperforming it are known for the multi-oracle setting. In this paper, we propose the state-wise maximum of the oracle policies' values as a natural baseline to resolve conflicting advice from multiple oracles. Using a reduction of policy optimization to online learning, we introduce a novel IL algorithm MAMBA, which can provably learn a policy competitive with this benchmark. In particular, MAMBA optimizes policies by using a gradient estimator in the style of generalized advantage estimation (GAE). Our theoretical analysis shows that this design makes MAMBA robust and enables it to outperform the oracle policies by a larger margin than the IL state of the art, even in the single-oracle case. In an evaluation against standard policy gradient with GAE and AggreVaTe(D), we showcase MAMBA's ability to leverage demonstrations both from a single and from multiple weak oracles, and significantly speed up policy optimization.

## 1 Introduction

Reinforcement learning (RL) promises to bring self-improving decision-making capability to many applications, including robotics [1], computer systems [2], recommender systems [3] and user interfaces [4]. However, deploying RL in any of these domains is fraught with numerous difficulties, as vanilla RL agents need to do a large amount of trial-and-error exploration before discovering good decision policies [5]. This inefficiency has motivated investigations into training RL agents with domain knowledge, an example of which is having access to oracle policies in the training phase.

The broad class of approaches that attempt to mimic or improve upon an available oracle policy is known as *imitation learning* (IL) [6]. Generally, IL algorithms work by invoking oracle policy demonstrations to guide an RL agent towards promising states and actions. As a result, oracle-level performance can be achieved without global exploration, thus avoiding RL's main source of high sample complexity. For IL with a single oracle policy, the oracle policy's return provides a natural benchmark for the agent to match or outperform. Most existing IL techniques assume this single-oracle setting, with a good *but possibly suboptimal* oracle policy. Behavior cloning [7] learns a policy from a fixed batch of trajectories in a supervised way by treating oracle actions as labels. Inverse reinforcement learning uses recorded oracle trajectories to infer the oracle's reward function [8–11]. Interactive IL [12, 13] assumes the learner can actively ask an oracle policy for a demonstration

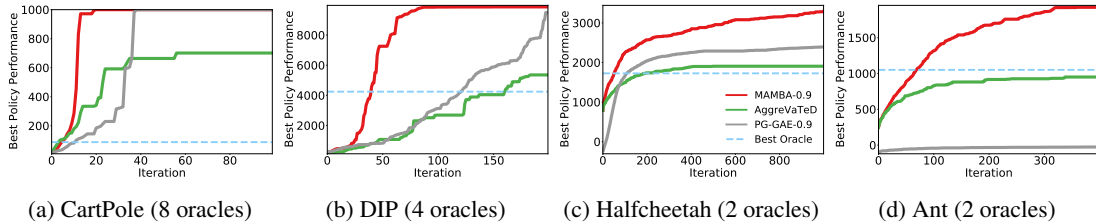

| (a) CartPole (8 oracles) | (b) DIP (4 oracles) | (c) Halfcheetah (2 oracles) | (d) Ant (2 oracles) |

Figure 1: Performance of the best policies returned by an RL algorithm (GAE policy gradient [17]), the single-oracle IL algorithm (AggreVaTeD [14]), and our multi-oracle IL algorithm MAMBA. All oracle polices here are suboptimal and AggreVaTeD imitates the best one. In Halfcheetah and Ant, policies of IL algorihtms are initialized by behavior cloning with the best oracle policy. A curve shows an algorithm's median performance across 8 random seeds. Please see Section 5 for details.

starting at the learner's current state. When reward information of the original RL problem is available, IL algorithms can outperform the oracle policy [14–16].

In this paper, we ask the question: how should an RL agent leverage domain knowledge encoded in *more than one* (potentially suboptimal) oracle policies? We study this question in the aforementioned interactive IL setting. Having multiple oracle policies is quite common in practice. For instance, consider the problem of minimizing task processing delays via load-balancing a network of compute nodes. Existing systems and their simulators have a number of human-designed heuristic policies for load balancing that can serve as oracles [18]. Likewise, in autonomous driving, available oracles can range from PID controllers to human drivers [19]. In these examples, each oracle has its own strengths and can provide desirable behaviors for different situations.

Intuitively, because more oracle policies can provide more information about the problem domain, an RL agent should be able to learn a good policy faster than using a single oracle. However, in reality, the agent does not know the properties of each oracle. What it sees instead are conflicting demonstrations from the oracle policies. Resolving this disagreement can be non-trivial, because there may not be a single oracle comprehensively outperforming the rest, and the quality of each oracle policy is unknown. Recently, several IL and RL works have started to study this practically important class of scenarios. InfoGAIL [20] conditions the learned policy on latent factors that motivate demonstrations of different oracles. AC-Teach [21] models each oracle with a set of attributes and relies on a Bayesian approach to decide which action to take based on their demonstrations. OIL [19] tries to identify and follow the best oracle in a given situation. However, all existing approaches to IL from multiple oracles sidestep two fundamental questions: (a) What is a reasonable benchmark for policy performance is these settings, analogous to the single-oracle policy quality in conventional IL? (b) Is there a systematic way to stitch together several suboptimal oracles into a stronger baseline that we can further improve upon?

We provide answers to these questions, making the following contributions:

1. We identify the state-wise maximum of oracle policies' values as a natural benchmark for learning from multiple oracles. We call it the *max-aggregated baseline* and propose policy improvement from it as a natural strategy to combine these oracles together, creating a new policy that is uniformly better than all the oracles in every state. These insights establish the missing theoretical foundation for designing algorithms for IL with multiple oracles.

2. We propose a novel IL algorithm called *MAMBA* (Max-aggregation of Multiple Baselines) to learn a policy that is competitive with the max-aggregated baseline by a reduction of policy optimization to online learning [13, 22]. MAMBA is a first-order algorithm based on a new IL gradient estimator designed in the spirit of generalized advantage estimation (GAE) [17] from the RL literature. Like some prior works in IL, MAMBA interacts with the oracle in a roll-in/roll-out fashion [13, 15] and does not assume access to oracle actions.

3. We provide regret-based performance guarantees for MAMBA. In short, MAMBA generalizes a popular single-oracle IL algorithm AggreVaTe(D) [13, 14] to learn from multiple oracles and to achieve larger improvements from suboptimal oracles. Empirically, we evaluate MAMBA against the IL baseline (AggreVaTeD [14]) and direct RL (GAE policy gradient [17]). Fig. 1 highlights the experimental results, where MAMBA demonstrates the capability to bootstrap demonstrations from multiple weak oracles to significantly speed up policy optimization.

## 2 Background: Episodic Interactive Imitation Learning

**Markov decision processes (MDPs)**  We consider finite-horizon MDPs with state space $\mathcal{S}$ and action space $\mathcal{A}$. Let $T$, $d_0(s)$, $\mathcal{P}(s'|s,a)$, and $r : \mathcal{S} \times \mathcal{A} \to [0,1]$ denote the problem horizon, the initial state distribution, the transition dynamics, and the reward function, respectively. *We assume that $d_0$, $\mathcal{P}$, and $r$ are fixed but unknown.* Given a class of state-dependent policies $\Pi$, our goal is to find a policy $\pi \in \Pi$ that maximizes the $T$-step return with respect to the initial state distribution $d_0$:

$$V^\pi(d_0) := \mathbb{E}_{s_0 \sim d_0} \mathbb{E}_{\xi_0 \sim \rho^\pi|s_0} \left[ \sum_{t=0}^{T-1} r(s_t, a_t) \right], \tag{1}$$

where $\rho^\pi(\xi_t|s_t)$ denotes the distribution over trajectory $\xi_t = s_t, a_t, \ldots, s_{T-1}, a_{T-1}$ generated by running policy $\pi$ starting from the state $s_t$ at time $t$ to the problem horizon. To compactly write down non-stationary processes, we structure the state space $\mathcal{S}$ as $\mathcal{S} = \bar{\mathcal{S}} \times \{0, T-1\}$, where $\bar{\mathcal{S}}$ is some basic state space; thus, $\mathcal{P}$ and $r$ can be non-stationary in $\bar{\mathcal{S}}$. We allow $\bar{\mathcal{S}}$ and $\mathcal{A}$ to be either discrete or continuous. We use the subscript of $t$ to emphasize the time index. When writing $s_t$ we assume it is at time $t$, and every transition from $s$ to $s'$ via $\mathcal{P}(s'|s,a)$ increments the time index by 1.

**State distributions and value functions**  We let $d_t^\pi$ stand for the state distribution at time $t$ induced by running policy $\pi$ starting from $d_0$ (i.e. $d_0^\pi = d_0$ for any $\pi$), and define the *average state distribution* as $d^\pi := \frac{1}{T} \sum_{t=0}^{T-1} d_t^\pi$. Sampling from $d^\pi$ returns $s_t$, where $t$ is uniformly distributed. Therefore, we can re-cast a policy's $T$-step return in (1) as $V^\pi(d_0) = T \mathbb{E}_{s \sim d^\pi} \mathbb{E}_{a \sim \pi|s}[r(s,a)]$. With a slight abuse of notation, we denote by $V^\pi : \mathcal{S} \to \mathbb{R}$ as the value function of policy $\pi$, which satisfies $V^\pi(d_0) = \mathbb{E}_{s \sim d_0}[V^\pi(s)]$. Given a function $f : \mathcal{S} \to \mathbb{R}$ such that $f(s_T) = 0$, we define the *Q-function w.r.t.* $f$ as $Q^f(s,a) := r(s,a) + \mathbb{E}_{s' \sim \mathcal{P}|s,a}[f(s')]$ and the *advantage function w.r.t.* $f$ as

$$A^f(s,a) := Q^f(s,a) - f(s) = r(s,a) + \mathbb{E}_{s' \sim \mathcal{P}|s,a}[f(s')] - f(s) \tag{2}$$

When $f = V^\pi$, we also write $A^{V^\pi} =: A^\pi$ and $Q^{V^\pi} =: Q^\pi$, which are the standard advantage and Q-functions of a policy $\pi$. We write $f$'s advantage function under a policy $\pi$ as $A^f(s,\pi) := \mathbb{E}_{a \sim \pi|s}[A^f(s,a)]$ and similarly $Q^f(s,\pi)$ and $f(d) := \mathbb{E}_{s \sim d}[f(s)]$ given a state distribution $d$. We refer to functions $f$ that index $Q$ or $A$ functions as *baseline value functions*, because we aim to improve upon the value they provide in each state.

**Definition 1.** We say a baseline value function $f$ is *improvable* w.r.t $\pi$ if $A^f(s,\pi) \geq 0, \forall s \in \mathcal{S}$.

**Policy optimization with multiple oracle policies**  The setup above describes a generic episodic RL problem, where the agent faces the need to perform strategic exploration and long-term credit assignment. A common approach to circumvent the exploration challenge in practice is by leveraging an oracle policy. In this paper, we assume access to *multiple* (potentially suboptimal) oracle policies during training, and leverage episodic interactive IL to improve upon them. We suppose that the learner (i.e. the agent) has access to a set of oracle policies $\Pi^e = \{\pi^k\}_{k \in [K]}$. During training, the learner can interact with the oracles in a roll-in-roll-out (RIRO) paradigm to collect demonstrations. In each episode, the learner starts from an initial state sampled from $d_0$ and runs its policy $\pi \in \Pi$ up to a switching time $t_e \in [0, T-1]$; then the learner asks an oracle policy $\pi^e \in \Pi^e$ to take over and finish the trajectory. At the end, the learner records the entire trajectory, including reward information. Note that we do not assume that oracle actions are observed. In addition, as sampled rewards are available here, the learner can potentially improve upon the oracle policies.

## 3 A Conceptual Framework for Learning from Multiple Oracles

In this paper, we focus on the scenario where the set $\Pi^e = \{\pi^k\}_{k \in [K]}$ contains more than one oracle policy. Having multiple oracle policies offers an opportunity to gain more information about the problem domain. Each oracle may be good in different situations, so the learner can query suitable oracles at different states for guidance. But how exactly can we leverage the information from multiple oracles to learn more efficiently than from any single one of them?

**Some natural baselines**  One approach for leveraging multiple oracles is to combine them into a single oracle, such as by using a fixed weighted mixture [23], or multiplying their action probabilities

in each state [24].[1] But the former can be quite bad even if only one oracle is bad, and the latter fails to combine two deterministic oracles. Another alternative is to evaluate each oracle and run a single-oracle IL algorithm with the one with the highest return. However, in Appendix A we show an example where two oracles have identical suboptimal returns, but switching between them results in the optimal behavior.

We ask, *is there a general principle for combining multiple oracles?* If we seek to switch among multiple oracles, how should switching points be chosen? Can we learn a rule for doing so reliably? In this section, we show that the issues mentioned above can be addressed by performing policy improvement upon the state-wise maximum over the oracles' values, i.e. the max-aggregated baseline. We describe two conceptual algorithms: one is based on the perfect knowledge of the MDP and the oracles' value functions, while the other builds on the first one using online learning to handle an unknown MDP and oracle values in the interactive IL setup. The insights gained from these two conceptual algorithms will be used to design their practical variation, MAMBA, in Section 4.

## 3.1 Max-aggregation with Policy Improvement

To illustrate the key idea, first let us suppose for a moment that perfect knowledge of the MDP and the oracles' value functions is available. In this idealized setting, the IL problem can be rephrased as follows: find a policy that is at least as good as all the oracle policies, and do so in a way whose computational complexity is *independent* of the problem horizon. The restriction on the complexity is important; otherwise we can just use the MDP knowledge to solve for the optimal policy.

How do we solve this idealized IL problem? When $\Pi^e$ contains only a single oracle, which we denote as $\pi^e$, a natural solution is the policy given by one-step policy improvement from $\pi^e$, i.e. the policy $\pi^e_+$ that acts according to $\arg\max_{a \in \mathcal{A}} r(s, a) + \mathbb{E}_{s' \sim \mathcal{P}|s,a}[V^{\pi^e}(s')]$. It is well known that this policy $\pi^e_+$ is uniformly better than $\pi^e$ for all the states, i.e. $V^{\pi^e_+}(s) \geq V^{\pi^e}(s)$ (cf. [25] and Corollary 1 below). However, this basic approach no longer applies when $\Pi^e$ contains multiple oracles, and natural attempts to invoke a single-oracle algorithm do not work in general as discussed earlier.

**A max following approach**   A simple way to remedy the failure mode of uniformly mixing the oracle policies is to take a non-uniform mixture that is aware of the quality of each oracle. *If* we have the value function of each oracle, we have a natural measure of their quality. With this intuition, for the $k$-th oracle policy $\pi^k \in \Pi^e$, let us write $V^k := V^{\pi^k}$. A natural candidate policy based on this idea is the greedy policy that follows the best oracle in any given state:

$$\pi^\bullet(a|s) := \pi^{k_s}(a|s), \qquad \text{where} \quad k_s := \arg\max_{k \in [K]} V^k(s) \tag{3}$$

Imitating a benchmark similar to $\pi^\bullet$ was recently proposed as a heuristic for IL with multiple oracles in [19]. Our first contribution is a theoretical result showing that the intuition behind this heuristic holds mathematically: $\pi^\bullet$ indeed satisfies $V^{\pi^\bullet}(s) \geq \max_{k \in [K]} V^k(s)$. To show this, we construct a helper corollary based on the useful Performance Difference Lemma (Lemma 1)[2].

**Lemma 1.** [26, 27] *Let $f : \mathcal{S} \to \mathbb{R}$ be such that $f(s_T) = 0$. For any MDP and policy $\pi$,*

$$V^\pi(d_0) - f(d_0) = T\mathbb{E}_{s \sim d^\pi}[A^f(s, \pi)]. \tag{4}$$

**Corollary 1.** *If $f$ is improvable w.r.t. $\pi$, then $V^\pi(s) \geq f(s), \forall s \in \mathcal{S}$.*

Corollary 1 implies that a policy $\pi$ has a better performance than all the oracles in $\Pi^e$ if there is a baseline value function $f$ that is improvable w.r.t. $\pi$ (i.e. $A^f(s, \pi) \geq 0$) and dominates the value functions of all oracle policies everywhere (i.e. $f(s) \geq V^k(s), \forall k \in [K], s \in \mathcal{S}$).

This observation suggests a natural value baseline for studying IL with multiple oracles:

$$f^{\max}(s) := \max_{k \in [K]} V^k(s). \tag{5}$$

Below we prove that this max-aggregated baseline $f^{\max}$ in (5) is improvable with respect to $\pi^\bullet$. Together with Corollary 1, this result implies that $\pi^\bullet$ is a valid solution to the idealized IL problem with multiple oracles. We write the advantage $A^{f^{\max}}$ with respect to $f^{\max}$ in short as $A^{\max}$.

**Proposition 1.** *$f^{\max}$ in (5) is improvable with respect to $\pi^\bullet$, i.e. $A^{\max}(s, \pi^\bullet) \geq 0$.*

**Degeneracy of max-following**  The policy $\pi^\bullet$ above, however, suffers from a degenerate case: when there is one oracle in $\Pi^e$ that is uniformly better than all the other oracles (say $\pi^e$), we have $\pi^\bullet = \pi^e$, whereas we know already $\pi^e_+$ is a uniformly better policy that we can construct using the same information. In this extreme case, in the standard IL setting with one oracle, $\pi^\bullet$ would simply return the suboptimal oracle.

**A max-*aggregation* approach**  Having noticed the failure mode of $\pi^\bullet$, we obtain a natural fix by combining the same value baseline (5) with the standard policy improvement operator. We define

$$\pi^{\max}(a|s) := \delta_{a=a_s}, \qquad \text{where} \qquad a_s := \arg\max_{a\in\mathcal{A}} A^{\max}(s,a), \tag{6}$$

and $\delta$ denotes the delta distribution. In contrast to $\pi^\bullet$, $\pi^{\max}$ looks one step ahead and takes the action with the largest advantage under $f^{\max}$. Note that this is *not* necessarily the same as following highest-value oracle in the current state. Since $\pi^{\max}$ satisfies $A^{\max}(s,\pi^{\max}) \geq A^{\max}(s,\pi^\bullet) \geq 0$, by Corollary 1, $\pi^{\max}$ is also a valid solution to the idealized IL problem with multiple oracles. The use of $\pi^{\max}$ is novel in IL to our knowledge, although $\pi^{\max}$ is called Multiple Path Construction Algorithm in controls [28, Chapter 6.4.2]. Corollary 1 provides a simple proof of why $\pi^{\max}$ works.

In general, $V^{\pi^{\max}}(s)$ and $V^{\pi^\bullet}(s)$ are not comparable. But, crucially, in the degenerate case above we see that $\pi^{\max}$ reduces to $\pi^e_+$ and therefore would perform better than $\pi^\bullet$, although in Appendix A we also show an MDP where $\pi^\bullet$ is better. Intuitively, this happens as $f^{\max}$ implicitly assumes using a single oracle for the remaining steps, but both $\pi^\bullet$ and $\pi^{\max}$ re-optimize their oracle choice at every step, so their relative quality can be arbitrary. While both $\pi^\bullet$ and $\pi^{\max}$ improve upon all the oracles, in this paper, we choose $\pi^{\max}$ as our imitation benchmark, because it is consistent with prior works in the single-oracle case and does not require observing the oracles' actions in IL, unlike $\pi^\bullet$.

Finally, we remark that Barreto et al. [29] recently proposed a generalized policy improvement operator, $\arg\max_{a\in\mathcal{A}} \max_{k\in[K]} Q^{\pi^k}(s,a)$, for multi-task RL problems. It yields a policy that is similar to $\pi^{\max}$ in (6) and is uniformly better than all the oracles. However, despite similarities, these two policies are overall different. In particular, $\max_{k\in[K]} Q^{\pi^k}(s,a) - f^{\max}(s) \leq A^{\max}(s,a)$, so (6) aims to improve upon a stronger baseline. And, as we will see in the next section, (6) allows for a geometric weighted generalization, which can lead to an even larger policy improvement.

### 3.2  Max-aggregation with Online Learning

The previous section shows that improving from the max-aggregated baseline $f^{\max}$ in (5) is a key to reconciling the conflicts between oracle policies. However doing so requires the knowledge of the MDP and the oracles' value functions, which are unavailable in the episodic interactive IL setting.

To compete with $f^{\max}$ without the above assumption, we design an IL algorithm via a reduction to online learning [30], a technique used in many prior works in the single-oracle setting [12–16, 31]. To highlight the main idea, at first we still assume that the oracles' value functions are given, but only the MDP is unknown. Then we show how to handle unknown value functions. For clarity, we use the subscript in $\pi_n$ to index the learner's policy in $\Pi$ generated in the $n$-th round of online learning, while using the superscript in $\pi^k$ to index the oracle policy in $\Pi^e$.

**Ideal setting with known values**  If the MDP dynamics and rewards are unknown, we can treat $d^{\pi_n}$ as the adversary in online learning and define the online loss in the $n$-th round as

$$\ell_n(\pi) := -T\mathbb{E}_{s\sim d^{\pi_n}}\left[A^{\max}(s,\pi)\right]. \tag{7}$$

By Lemma 1, making $\ell_n(\pi_n)$ small ensures that $V^{\pi_n}(d_0)$ is not much worse than $f^{\max}(d_0)$. Formally, averaging this argument over $N$ rounds of online learning, we obtain

$$\frac{1}{N}\sum_{n\in[N]} V^{\pi_n}(d_0) = f^{\max}(d_0) + \Delta_N - \epsilon_N(\Pi) - \frac{\text{Regret}_N}{N}, \tag{8}$$

where we define $\text{Regret}_N := \sum_{n=1}^{N} \ell_n(\pi_n) - \min_{\pi\in\Pi}\sum_{n=1}^{N}\ell_n(\pi)$,

$$\Delta_N := \frac{-1}{N}\sum_{n=1}^{N}\ell_n(\pi^{\max}), \text{ and } \epsilon_N(\Pi) := \min_{\pi\in\Pi}\frac{1}{N}\left(\sum_{n=1}^{N}\ell_n(\pi) - \sum_{n=1}^{N}\ell_n(\pi^{\max})\right). \tag{9}$$

In (8), the regret characterizes the learning speed of an online algorithm, while $\epsilon_N(\Pi)$ captures the quality of the policy class. If $\pi^{\max} \in \Pi$, then $\epsilon_N(\Pi) = 0$; otherwise, $\epsilon_N(\Pi) \geq 0$. Furthermore, we have $\Delta_N \geq 0$, because we showed $A^{\max}(s,\pi^{\max}) \geq 0$ in Section 3.1. Thus,

---

**Algorithm 1** MAMBA for IL with multiple oracles

---

**Input:** Initial learner policy $\pi_1$, oracle polices $\{\pi^k\}_{k\in[K]}$, function approximators $\{\widehat{V}^k\}_{k\in[K]}$.
**Output:** The best policy in $\{\pi_1, \ldots, \pi_N\}$.
 1: **for** $n = 1 \ldots N-1$ **do**
 2:    Uniformly sample $t_e \in [T-1]$ and $k \in [K]$.
 3:    Roll-in $\pi_n$ up to $t_e$ and switch to $\pi^k$ to complete the remaining trajectory to collect data $\mathcal{D}_n$.
 4:    Update $\widehat{V}^k$ using $\mathcal{D}_n$ (e.g. using Monte-Carlo estimates).
 5:    Roll-in $\pi_n$ for the full $T$-horizon to collect data $\mathcal{D}'_n$.
 6:    Compute the sample estimate $g_n$ of $\nabla \widehat{\ell}_n(\pi; \lambda)$ (17) using $\mathcal{D}'_n$ and $\widehat{f}^{\max}(s) = \max_{k\in[K]} \widehat{V}^k(s)$.
 7:    Update $\pi_n$ to $\pi_{n+1}$ by giving $g_n$ to a first-order online learning algorithm (e.g. mirror descent).
 8: **end for**

---

when $\pi^{\max} \in \Pi$, running a no-regret algorithm (i.e. an algorithm such that $\text{Regret}_N = o(N)$) to solve this online learning problem will guarantee producing a policy whose performance at least $\mathbb{E}_{s\sim d_0}[\max_{k\in[K]} V^k(s)] + \Delta_N + o(1)$ after $N$ rounds.

The above reduction in (7) generalizes AggreVaTE [13] from using $f = V^{\pi^e}$ in $A^f$ for define the online loss for the single oracle case to using $f = f^{\max}$ instead, which is also applicable to multiple oracles. When an oracle in $\Pi^e$ dominates the others for all the states, (7) is the same as the online loss in AggreVaTE.

**Effect of approximate oracle values**   Recall that for the above derivation we assumed oracle policy values (and hence $f^{\max}$) are given. In practice, $f^{\max}$ is unavailable and needs to be approximated by some $\widehat{f}^{\max}$. Let $\widehat{A}$ denote the shorthand of $A^{\widehat{f}^{\max}}$. We can treat the approximation error as bias and variance in the feedback signal, such as the sample estimate of the gradient below:

$$\nabla \widehat{\ell}_n(\pi_n) = -T\mathbb{E}_{s\sim d^{\pi_n}}\mathbb{E}_{a\sim\pi|s}\left[\nabla \log \pi(a|s)\widehat{A}(s,a)\right]|_{\pi=\pi_n}, \tag{10}$$

where $\nabla$ is with respect to the policy. We summarize the approximation effects as a meta theorem, where generally $\beta$ and $\nu$ increase as the problem horizon increases.

**Theorem 1.** *Suppose a first-order online algorithm that satisfies $\mathbb{E}[\text{Regret}_N] \leq O(\beta N + \sqrt{\nu N})$ is adopted, where $\beta$ and $\nu$ are the bias and the variance of the gradient estimates, respectively. Then*

$$\mathbb{E}[\max_{n\in[N]} V^{\pi_n}(d_0)] \geq \mathbb{E}_{s\sim d_0}[\max_{k\in[K]} V^k(s)] + \mathbb{E}[\Delta_N - \epsilon_N(\Pi)] - O(\beta + \sqrt{\nu}N^{-1/2}) \tag{11}$$

*where the expectation is over the randomness in feedback and the online algorithm.*

Theorem 1 describes considerations of using $\widehat{f}^{\max}$ in place of $f^{\max}$. For the single-oracle case, $\widehat{f}^{\max}$ can be an unbiased Monte-Carlo estimate (i.e. $\nabla \widehat{\ell}_n = \nabla \ell_n$); but a sample estimate of such $\nabla \widehat{\ell}_n(\pi_n)$ suffers from a variance that is $T$-times larger than the Monte-Carlo estimate of policy gradient due to the restriction of RIRO data collection protocol[3]. Alternatively, one can use function approximators [14] as $\widehat{f}^{\max}$ to shift the variance from the gradient estimate to learning $\widehat{f}^{\max}$. In this case, (10) becomes akin to the actor-critic policy gradient. But when the accuracy of the value estimate $\widehat{f}^{\max}$ is bad, the bias in (10) can also compromise the policy learning.

For the multi-oracle case, unbiased Monte-Carlo estimates of $f^{\max}$ are infeasible, because $f^{\max}(s) = V^{k_s}(s)$ and $\pi^{k_s}$ is unknown (i.e. we do not know the best oracle policy at state $s$). Therefore, $\widehat{f}^{\max}$ in (10) must be a function approximator. But, due to the max operator in $f^{\max}$, learning $\widehat{f}^{\max}$ becomes challenging as all the oracles' value functions need to ba approximated uniformly well.

## 4   MAMBA: Multi-Step Policy Improvement upon Multiple Oracles

We propose MAMBA as a practical realization of the first-order reduction idea in Theorem 1 (shown in Algorithm 1). As discussed, obtaining a good sample estimate of (10) is nontrivial. As a workaround,

we will design MAMBA based on an alternate online loss $\ell_n(\pi; \lambda)$ that shares the same property as $\ell_n(\pi)$ in (7) but has a gradient expression with tunable bias-variance trade-off. More importantly, we prove that adapting $\lambda$ leads to a continuous transition from one-step policy improvement to solving a full-scale RL problem. Consequently, fine-tuning this $\lambda$ "knob" can additionally trade off the limited performance gain of performing one-step policy improvement and the sample inefficiency of solving a full-scale RL problem in order to obtain the best finite-sample policy performance in learning.

## 4.1 Trade-off between One-Step Policy Improvement and Full RL

The alternate online loss $\ell_n(\pi; \lambda)$ is based on the geoemtric weighting technique commonly used in RL algorithms (such as TD-$\lambda$ [32]). Specifically, for $\lambda \in [0, 1]$, we propose to define the new online loss in the $n$-th round as

$$\ell_n(\pi; \lambda) := -(1 - \lambda)T\mathbb{E}_{s \sim d^{\pi_n}}\left[A_\lambda^{\max,\pi}(s, \pi)\right] - \lambda\mathbb{E}_{s \sim d_0}\left[A_\lambda^{\max,\pi}(s, \pi)\right] \tag{12}$$

where we define a $\lambda$-weighted advantage

$$A_\lambda^{\max,\pi}(s, a) := (1 - \lambda)\sum_{i=0}^\infty \lambda^i A_{(i)}^{\max,\pi}(s, a) \tag{13}$$

by combining various $i$-step advantages:

$$A_{(i)}^{\max,\pi}(s_t, a_t) := \mathbb{E}_{\xi_t \sim \rho^\pi|s_t}[r(s_t, a_t) + \cdots + r(s_{t+i}, a_{t+i}) + f^{\max}(s_{t+i+1})] - f^{\max}(s_t).$$

The hyperparameter $\lambda$ in (12) controls the attainable policy performance and the sample complexity of learning with (12). To examine this, let us first note some identities due to the geometric weighting: $A_0^{\max,\pi} = A_{(0)}^{\max,\pi} = A^{\max}$ and $A_1^{\max,\pi} = A_{(\infty)}^{\max,\pi} = Q^\pi - f^{\max}$. Therefore, when $\lambda = 1$, the online loss in (12) becomes the original RL objective for every round (one can show that $\ell_n(\pi; 1) = f^{\max}(d_0) - V^\pi(d_0)$ for all $n$), where policy learning aims to discover the optimal policy but is sample inefficient. On the other hand, when $\lambda = 0$ (we define $0^0 = 1$), (12) reduces back to (7) (i.e., $\ell_n(\pi; 0) = \ell_n(\pi)$), where policy learning is about one-step policy improvement from the max-aggregated baseline (namely IL). Although doing so does not necessarily yield the optimal policy, it can be done in a sample-efficient manner. By tuning $\lambda$ we can trade off performance bias and sample complexity to achieve the best performance given a *finite* number of samples.

We can make a connection of (12) to the IL literature. When there is a single oracle (i.e. $f^{\max} = V^{\pi^e}$), we can interpret the online loss (12) in terms of known IL objectives. In (12), the first term is the $\lambda$-weighted version of the AggreVaTe loss [13], and the second term is the $\lambda$-weighted version of the THOR loss [33]. But none of these prior IL algorithms makes use of the $\lambda$-weighted advantages.

Lastly, let us formally establish the relationship between (12) and the RL objective by a generalization of the Performance Difference Lemma (Lemma 1) to take into account geometric weighting.

**Lemma 2.** *For any policy $\pi$, any $\lambda \in [0, 1]$, and any baseline value function $f : \mathcal{S} \to \mathbb{R}$,*

$$V^\pi(d_0) - f(d_0) = (1 - \lambda)T\mathbb{E}_{s \sim d^\pi}\left[A_\lambda^{f,\pi}(s, \pi)\right] + \lambda\mathbb{E}_{s \sim d_0}\left[A_\lambda^{f,\pi}(s, \pi)\right], \tag{14}$$

*where $A_\lambda^{f,\pi}$ is defined like $A_\lambda^{\max,\pi}$ but with a general $f$ instead of $f^{\max}$.*

In other words, the new online loss function $\ell_n(\pi; \lambda)$ satisfies an equality similar to Lemma 1, which $\ell_n(\pi)$ relies on to establish the reduction in (8). Now, with the proper generalization given by Lemma 2, we can formally justify learning with $\ell_n(\pi_n; \lambda)$.

**Theorem 2.** *Performing no-regret online learning w.r.t.* (12) *has the guarantee in Theorem 1, except now $\epsilon_N(\Pi)$ can be negative when $\lambda > 0$.*

Theorem 2 shows that learning with $\ell_n(\pi; \lambda)$ has a similar performance guarantee to using $\ell_n(\pi)$ in Theorem 1, but with one important exception: now $\epsilon_N(\Pi)$ can be negative (which is in our favor), because $\pi^{\max}$ may not be the best policy for the multi-step advantage in $\ell_n(\pi; \lambda)$ when $\lambda > 0$. This again can be seen from the fact that optimizing $\ell_n(\pi; 1)$ is equivalent to direct RL. As a result, when using larger $\lambda$ in MAMBA, larger improvements can be made from the oracle polices as we move from one-step policy improvement toward full-fledged RL. However, looking multiple steps ahead with high $\lambda$ would also increase the feedback variance $\nu$ in Theorem 1, which results in a slower convergence rate. In practice, $\lambda$ needs to be tuned to achieve the best finite-sample performance.

## 4.2 Simple Gradient Estimation

While the online loss $\ell_n(\pi; \lambda)$ in (12) appears complicated, interestingly, its gradient $\nabla \ell_n(\pi; \lambda)$ has a very clean expression, as we prove below.

**Lemma 3.** *For any $\lambda \in [0, 1]$, any baseline value function $f : \mathcal{S} \to \mathbb{R}$, and any policy $\pi$, the following holds:*

$$h(\pi; \lambda) := (1 - \lambda) T \mathbb{E}_{s \sim d^\mu} \left[ A_\lambda^{f,\pi}(s, \pi) \right] + \lambda \mathbb{E}_{s \sim d_0} \left[ A_\lambda^{f,\pi}(s, \pi) \right] \tag{15}$$

$$\nabla h(\pi; \lambda)|_{\mu=\pi} = T \mathbb{E}_{s \sim d^\pi} \mathbb{E}_{a \sim \pi|s}[\nabla \log \pi(a|s) A_\lambda^{f,\pi}(s, a)], \tag{16}$$

*where $d^\mu$ denotes the average state distribution of a policy $\mu$.*

Using Lemma 3, we design a gradient estimator for $\ell_n(\pi; \lambda)$ by approximating $f^{\max}$ in $\nabla \ell_n(\pi; \lambda)$ with a function approximator $\widehat{f}^{\max}$:

$$\nabla \widehat{\ell}_n(\pi_n; \lambda) = -T \mathbb{E}_{s \sim d^{\pi_n}} \mathbb{E}_{a \sim \pi|s}[\nabla \log \pi(a|s) \widehat{A}_\lambda^\pi(s, a)]|_{\pi=\pi_n}, \tag{17}$$

where $\widehat{A}_\lambda$ is defined by replacing $f^{\max}$ in (13) with $\widehat{f}^{\max}$. Implementation-wise, the infinite sum in $\widehat{A}_\lambda$ can be computed by standard dynamic programming.

**Lemma 4.** *Define $\widehat{A}(s, a) := r(s, a) + \mathbb{E}_{s'|s,a}[\widehat{f}^{\max}(s')] - \widehat{f}^{\max}(s)$. It holds that for all $\lambda \in [0, 1]$,*

$$\widehat{A}_\lambda^\pi(s_t, a_t) = \mathbb{E}_{\xi_t \sim \rho^\pi|s_t} \left[ \sum_{\tau=t}^{T-1} \lambda^{\tau-t} \widehat{A}(a_\tau, s_\tau) \right] \tag{18}$$

Therefore, given $\widehat{f}^{\max}$ as a function approximator, an unbiased estimate of (17) can be obtained by sampling trajectories from $\pi$ directly, without any RIRO interleaving with the oracles. This gradient estimator is reminiscent of GAE for policy gradient [17], but translated to IL.

The gradient expression in (17) reveals that $\lambda$ also plays a role in terms of bias-variance trade-off, in addition to the transition from one-step improvement to full-scale RL discussed above. Comparing (10) and (17), we see that $\nabla \widehat{\ell}_n(\pi; \lambda)$ in (17) replaces $\widehat{A}$ in $\nabla \widehat{\ell}_n(\pi)$ in (10) with the $\lambda$-weighted version $\widehat{A}_\lambda^\pi$. Controlling $\lambda$ regulates the effects of the approximation error $\widehat{f}^{\max} - f^{\max}$ on the difference $\widehat{A}_\lambda^\pi - A_\lambda^{\max,\pi}$, which in turn determines the gradient bias $\nabla \widehat{\ell}_n(\pi; \lambda) - \nabla \ell_n(\pi; \lambda)$ (namely $\beta$ in Theorem 1). This mechanism is similar to the properties of the GAE policy gradient [17]. We recover the gradient in (10) when $\lambda = 0$ and the policy gradient when $\lambda = 1$.

Finally, we emphasize that $\nabla \widehat{\ell}_n(\pi; \lambda)$ in (17) is *not* an approximation of $\nabla \widehat{\ell}_n(\pi)$ in (10), because generally $\nabla \ell_n(\pi_n; \lambda) \neq \nabla \ell_n(\pi_n)$ even when $\widehat{f}^{\max} = f^{\max}$, except for $f^{\max} = V^{\pi_n}$ (in this case, for all $\lambda \in [0, 1]$, $\nabla \ell_n(\pi_n; \lambda) = \nabla \ell_n(\pi_n) = -\nabla V^{\pi_n}(d_0)$, which is the negative policy gradient). Therefore, while the GAE policy gradient [17] is an approximation of the policy gradient, $\nabla \widehat{\ell}_n(\pi)$ and $\nabla \widehat{\ell}_n(\pi; \lambda)$ are gradient approximations of *different* online loss functions in (7) and (12).

## 5 Experiments and Discussion

We corroborate our theoretical discoveries with simulations of IL from multiple oracles. We compare MAMBA with two representative algorithms: GAE Policy Gradient [17] (PG-GAE with $\lambda = 0.9$) for direct RL and AggreVaTeD [14] for IL with a single oracle. Because we can view these algorithms as different first-order feedback for policy optimization, comparing their performance allows us to study two important questions: 1) whether the proposed GAE-style gradient in (17) is an effective update direction for IL and 2) whether using multiple oracles helps the agent learn faster.

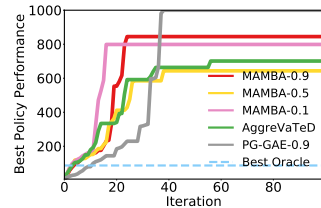

Figure 2: MAMBA with different $\lambda$ values in a single-oracle setup in CartPole. A curve shows an algorithm's median performance across 8 random seeds.

Four continuous Gym [34] environments are used: CartPole and DoubleInvertedPendulum (DIP) based on DART physics engine [35], and Halfcheetah and Ant based on Mujoco physics engine [36]. To facilitate a meaningful comparison, we let these three algorithms use the same first-order optimizer[4], train the same initial neural network policies, and share the same

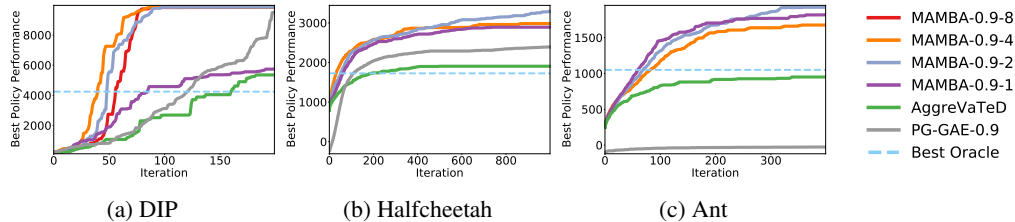

(a) DIP           (b) Halfcheetah           (c) Ant

Figure 3: Comparison of MAMBA with different number of oracles ($\lambda = 0.9$). A curve shows an algorithm's median performance across 8 random seeds.

random seeds. In each training iteration, an algorithm would perform $H$ rollouts following the RIRO paradigm (see also Algorithm 1), where $H = 8$ for CartPole and DIP and $H = 256$ for Halfcheetah and Ant. Each oracle policy here is a partially trained, *suboptimal* neural network[5], and its value function approximator used in MAMBA is trained online along policy optimization, by Monte-Carlo estimates obtained from the RIRO data collection. Please see Appendix D for implementation details. The codes are provided at `https://github.com/microsoft/MAMBA`.

**Effects of $\lambda$-weighting**    First we consider in Fig. 2 the *single*-oracle IL setting with the oracle that has the highest return in the CartPole environment. We see that with the help of the oracle policy, AggreVaTeD (which is MAMBA with $\lambda = 0$) improves faster than PG-GAE. However, while AggreVaTeD learns to significantly outperform the oracle, it does not reach the optimal performance, like PG-GAE. To improve the learner performance, we use MAMBA with $\lambda > 0$ to learn from the same suboptimal oracle. As shown in Theorem 2 using a positive $\lambda$ can increase the amount of improvement that can be made from the oracle compared with $\lambda = 0$. The trend in Fig. 2 supports this insight, where high $\lambda = 0.9$ allows the learner to reach higher return. We found that using the middle $\lambda = 0.5$ gives the worst performance, likely because it settles badly in the trade-off of properties. In the following experiments, we will use $\lambda = 0.9$ for MAMBA as it performs the best here.

**Effects of multiple oracles**    We show the effects of using multiple oracles in Fig. 3 for the remaining three environments (the results of CartPole can be found in Fig. 1 and Appendix D). Here we run MAMBA with $\lambda = 0.9$ with 1, 2, 4, or 8 oracles. We index these oracles in a descending order of their performances with respect to the initial state distribution; e.g., MAMBA-0.9-1 uses the best oracle and MAMBA-0.9-2 uses the top two. Interestingly, by including more but *strictly weaker* oracles, MAMBA starts to improve the performance of policy optimization. Overall MAMBA greatly outperforms PG-GAE and AggreVaTeD across all the environments, even when just using a single oracle.[6] The benefit of using more oracles becomes smaller, however, as we move to the right of Fig. 3 into higher dimensional problems. Although using more oracles can potentially yield higher performance, the learner also needs to spend more time to learn the oracles' value functions. Therefore, under the same interaction budget, the value approximation quality worsens when there are more oracles. This phenomenon manifests particularly in our direct implementation, which trains each value function estimator independently with Monte-Carlo samples. We believe that the scalability for dimensionality can be improved by using off-policy techniques and sharing parameters across different networks. We leave this as an important future direction.

**Summary**    We conclude this paper by revisiting Fig. 1, which showcases the best multi-oracle settings in Fig. 3. Overall these results support the benefits of IL from multiple oracles and the new GAE-style IL gradient in (17). In conclusion, we propose a novel theoretical foundation and algorithm MAMBA for IL with multiple oracle policies. We study how the conflicts between different oracles can be resolved through the max-aggregated baseline and propose a new GAE-style gradient for the IL setting, which can also be used to improve the robustness and performance of existing single-oracle IL algorithms. We provide regret-based theoretical guarantees on MAMBA and demonstrate its properties empirically. The experimental results show that MAMBA is able to improve upon multiple, very suboptimal oracle policies to achieve the optimal performance, faster than both the pure RL method (PG-GAE [17]) and the single-oracle IL algorithm (AggreVaTeD [14]).

## Broader Impact

This paper is theoretical in nature, and so we expect the ethical and societal consequences of our specific results to be minimal. More broadly, we do expect that reinforcement learning will have significant impact on society. There is much potential for benefits to humanity in the often-referenced application domains of precision medicine, personalized education, and elsewhere. There is also much potential for harms, both malicious and unintentional. To this end, we hope that research into the foundations of reinforcement learning can help enable these applications and mitigate harms through the development of algorithms that are efficient, robust, and safe.

## Footnotes

[1]These approaches are proposed for supervised learning and not specifically IL.

[2]Lemma 1 is an adaptation of the standard Performance Difference Lemma to using $f$ that is not necessarily the value function of any policy. We provide proofs of Lemma 1 and Corollary 1 in Appendix C for completeness.

[3]As $V^{k_s}$ is not the value of $\pi_n$ but $\pi^{k_s}$, computing an unbiased estimate of $\nabla \ell_n(\pi_n)$ requires uniformly selecting the switching time $t_e \in \{0, \ldots, T-1\}$ in the RIRO setting, which amplifies the variance by $O(T)$.

[4]ADAM [37] for CartPole, and Natural Gradient Descent [38] for DIP, Halfcheetah, and Ant

[5]The best and the worst oracles in scores of 87 and 9 in CartPole, 4244 and 2440 in DIP, 1726 and 1395 in Halfcheetah, and 1051 and 776 in Ant.

[6]The bad performance of PG-GAE in Ant is due to that all learners can only collect a fixed number of trajectories in each iteration, as opposed to a fixed number of samples used in usual RL benchmarks. This setting is a harder RL problem and better resembles real-world data collection where reset is expensive.

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
