[Supplementary Material]

# Appendix

## A  MDP Examples for Section 3

**Problem with selecting oracles based on initial value.**   In the example of Figure 4, each oracle $\pi_\ell$ and $\pi_r$ individually gets same the suboptimal reward of $1/2$. Alternatively, we can switch between the oracles once to get a reward of $3/4$, and twice to get the optimal reward of $1$.

Figure 4: An example MDP for illustrative purposes. All terminal states not shown give a reward of $0$ and intermediate states have no rewards. Two oracle policies $\pi_\ell$ and $\pi_r$ choose the left and right actions respectively in each state. The optimal terminal state is outlined in bold.

**Ordering of $\pi^\bullet$ and $\pi^{\max}$.**   Consider the example MDP of Figure 5. In the state $s_0$, the policy $\pi^\bullet$ selects the oracle with largest value in $s_0$ and goes left. It subsequently selects the right oracle in $s_1$ and left in $s_4$ to get the optimal reward. $\pi^{\max}$ on the other hand chooses between the left and right actions in $s_0$ based on $f^{\max}(s_1) = 0.7$ and $f^{\max}(s_2) = 3/4$. Consequently it goes right and eventually obtains a suboptimal reward of $3/4$. In this case, we see that $\pi^\bullet$ is better than $\pi^{\max}$. On the other hand, if we swap the rewards of $s_7$ and $s_{11}$, then $\pi^\bullet$ chooses the right action in $s_0$ and gets a suboptimal reward. Further swapping the rewards of $s_9$ and $s_{10}$ makes $\pi^{\max}$ pick the left action in $s_0$ and it eventually reaches the optimal reward. This illustrates clearly that $\pi^{\max}$ and $\pi^\bullet$ are incomparable in general.

Figure 5: An alternate example MDP for illustrative purposes. All terminal states not shown give a reward of $0$ and intermediate states have no rewards. Two oracle policies $\pi_\ell$ and $\pi_r$ choose the left and right actions respectively in each state. The optimal terminal state is outlined in bold.

# B  Additional Notes on Related Work

Several prior works proposed empirical approaches to IL settings with multiple oracles. Info-GAIL [20] is an extension of GAIL [39] that aims at automatically identifying semantically meaningful latent factors that can explain variations in demonstrations across oracles. It assumes that demonstrations come from a mixture-of-oracles policy, where each demonstration is generated by sampling a value of the latent factors from a prior and using it to condition an oracle's action choices. InfoGAIL tries to recover this oracle mixture. In contrast, MAMBA can be viewed as choosing actions based on learned estimates of oracles' value functions without imitating any single oracle or their mixture directly. In multi-modal IL [40], latent factors conceptually similar to those in InfoGAIL correspond to different skills being demonstrated by the oracles, and Tamar et al. [41]'s approach focuses on settings where these factors characterize different oracles' intentions. OIL [19] is more similar to MAMBA: like MAMBA, it uses individual oracle policies' state values to decide on an action in a given state. However, OIL does so by using the best-performing oracle in a given state as the learner's "critic" and doesn't justify its approach theoretically.

At least two algorithms have used a Bayesian approach to decide which oracle to trust in a multiple-oracle setting. AC-Teach [21] models each oracle with a set of attributes and relies on a Bayesian approach to decide which action to take based on their demonstrations. Gimelfarb et al. [42] assumes that oracles propose reward functions, some of which are inaccurate, and uses Bayesian model combination to aggregate their advice. Out of the two, AC-Teach can be regarded as a Bayesian counterpart of MAMBA, but, like other existing approaches to learning from multiple oracles, doesn't come with a theoretical analysis.

Lastly, we remark that the improvement made from the suboptimal oracles in MAMBA is attributed to using reward information in IL, similar to AggreVaTeD [14] but different from behavior cloning [7] or DAgger [12]. While we allow weaker oracle policies than are typical in IL literature, they still need to have meaningful behaviors to provide an informative advantage MAMBA to improve upon; e.g., they cannot be completely uniformly random policies as is often done in the batch RL setting.

# C  Proofs

## C.1  Proof of Lemma 1

**Lemma 1.** [26, 27] *Let* $f : \mathcal{S} \to \mathbb{R}$ *be such that* $f(s_T) = 0$. *For any MDP and policy* $\pi$,

$$V^\pi(d_0) - f(d_0) = T\mathbb{E}_{s \sim d^\pi}[A^f(s, \pi)]. \tag{4}$$

*Proof.* By definition of $d^\pi$, we can write

$$V^\pi(d_0) = T\mathbb{E}_{s \sim d^\pi}\mathbb{E}_{s \sim \pi|s}[r(s, a)] = T\mathbb{E}_{s \sim d^\pi}[r(s, \pi)]$$

On the other hand, we can write

$$-f(d_0) = \sum_{t=1}^{T-1} f(d_t) - \sum_{t=0}^{T-1} f(d_t) = T\mathbb{E}_{s \sim d^\pi}[\mathbb{E}_{a \sim \pi|s}\mathbb{E}_{s' \sim \mathcal{P}|s,a}[f(s')] - f(s)]$$

Combing the two equalities shows the result. ∎

## C.2  Proof of Corollary 1

**Corollary 1.** *If* $f$ *is improvable w.r.t.* $\pi$, *then* $V^\pi(s) \geq f(s)$, $\forall s \in \mathcal{S}$.

*Proof.* Because Lemma 1 holds for any MDP, given the state $s$ in Corollary 1 we can define a new MDP whose initial state is at $s$ and properly adapt the problem horizon. Then Corollary 1 follows directly from applying Lemma 1 to this new MDP. ∎

## C.3  Proof of Proposition 1

**Proposition 1.** $f^{\max}$ *in* (5) *is improvable with respect to* $\pi^\bullet$, *i.e.* $A^{\max}(s, \pi^\bullet) \geq 0$.

*Proof.* Let us recall the definition (3) of $k_s$ and let us assume without loss of generality that $k_s = 1$. We observe that

$$A^{\max}(s, \pi^{\bullet}) = r(s, \pi^{\bullet}) + \mathbb{E}_{a \sim \pi^{\bullet}|s} \mathbb{E}_{s' \sim \mathcal{P}|s,a}[f^{\max}(s')] - f^{\max}(s)$$

$$\geq r(s, \pi^{\bullet}) + \mathbb{E}_{a \sim \pi^{\bullet}|s} \mathbb{E}_{s' \sim \mathcal{P}|s,a}[V^1(s')] - V^1(s) = A^{V^1}(s, \pi^1) \geq 0$$

where the last step follows since $\pi^{\bullet}(a|s) = \pi^{k_s}(a|s) = \pi^1(a|s)$ and the advantage of a policy with respect to its value function is always 0. ∎

## C.4   Proof of Theorem 1

**Theorem 1.** *Suppose a first-order online algorithm that satisfies $\mathbb{E}[Regret_N] \leq O(\beta N + \sqrt{\nu N})$ is adopted, where $\beta$ and $\nu$ are the bias and the variance of the gradient estimates, respectively. Then*

$$\mathbb{E}[\max_{n \in [N]} V^{\pi_n}(d_0)] \geq \mathbb{E}_{s \sim d_0}[\max_{k \in [K]} V^k(s)] + \mathbb{E}[\Delta_N - \epsilon_N(\Pi)] - O(\beta + \sqrt{\nu} N^{-1/2}) \qquad (11)$$

*where the expectation is over the randomness in feedback and the online algorithm.*

*Proof.* By using (8) and the assumption on the first-order algorithm, we can write

$$\mathbb{E}\left[\frac{1}{N} \sum_{n \in [N]} V^{\pi_n}(d_0)\right] = f^{\max}(d_0) + \mathbb{E}\left[\Delta_N - \epsilon_N(\Pi) - \frac{Regret_N}{N}\right]$$

$$\geq f^{\max}(d_0) + \mathbb{E}[\Delta_N - \epsilon_N(\Pi)] - O\left(\beta + \sqrt{\frac{\nu}{N}}\right)$$

Finally, using $\frac{1}{N} \sum_{n \in [N]} V^{\pi_n}(d_0) \leq \max_{n \in [N]} V^{\pi_n}(d_0)$ and the definition of $f^{\max}$, we have the final statement. ∎

## C.5   Proof of Lemma 2

**Lemma 2.** *For any policy $\pi$, any $\lambda \in [0, 1]$, and any baseline value function $f : \mathcal{S} \to \mathbb{R}$,*

$$V^\pi(d_0) - f(d_0) = (1 - \lambda) T \mathbb{E}_{s \sim d^\pi}\left[A_\lambda^{f,\pi}(s, \pi)\right] + \lambda \mathbb{E}_{s \sim d_0}\left[A_\lambda^{f,\pi}(s, \pi)\right], \qquad (14)$$

*where $A_\lambda^{f,\pi}$ is defined like $A_\lambda^{\max,\pi}$ but with a general $f$ instead of $f^{\max}$.*

*Proof.* The proof is uses a new generalization of the Performance Difference Lemma (Lemma 1), which we state generally for non-Markovian processes. A similar equality holds for the infinite-horizon discounted problems.

**Lemma 5** (Non-even performance difference lemma)**.** *Let $\pi$ be a policy and let $f$ be any function that is history dependent such that $\mathbb{E}_{h_T \sim d_T^\pi}[f(h_T)] = 0$. Let $\tau_0, \tau_1, \tau_2, \ldots \tau_I$ be monotonically increasing integers where $\tau_0 = 0$ and $\tau_I = T$. For any non-Markovian decision process, it holds that,*

$$V^\pi(d_0) - f(d_0) = \sum_{k=0}^{I-1} \mathbb{E}_{h_{\tau_k} \sim d_{\tau_k}^\pi}[A_{(i_k)}^{f,\pi}(h_{\tau_k}, \pi)]$$

*where $i_k = \tau_{k+1} - \tau_k - 1$.*

*Proof of Lemma 5.* By definition,

$$V^\pi(d_0) = \sum_{t=0}^{T-1} \mathbb{E}_{h_t \sim d_t^\pi}[r(h_t, \pi)] = \sum_{k=0}^{I-1} \mathbb{E}_{h_{\tau_k} \sim d_{\tau_k}^\pi} \mathbb{E}_{\rho^\pi|h_{\tau_k}}\left[\sum_{t=\tau_k}^{\tau_{k+1}-1} r(h_t, a_t)\right]$$

On the other hand, we can write $-f(d_0) = \sum_{k=1}^{I} f(d_{\tau_k}^\pi) - \sum_{k=0}^{I-1} f(d_{\tau_k}^\pi)$. Combing the two equalities shows the result. ∎

Now return to the Markovian case. Using Lemma 5, we derive a $\lambda$-weighted Performance Difference Lemma (Lemma 2). A history dependent (discounted) version can be shown similarly. To simplify writing, we let $\Theta = V^\pi(d_0) - f(d_0)$ and $A_{(i)} = A_{(i)}^{f,\pi}$ as shorthands, and we will omit the dependency on random variables in the expectation. Using Lemma 5, we can write

$$\Theta = \sum_{t=0,1,\ldots,T-1} \mathbb{E}_{d_t^\pi} \mathbb{E}_\pi [A_{(0)}]$$

$$2\Theta = \sum_{t=0,2,4\ldots} \mathbb{E}_{d_t^\pi} \mathbb{E}_\pi [A_{(1)}] + \left( \mathbb{E}_{d_0} \mathbb{E}_\pi [A_{(0)}] + \sum_{t=1,3\ldots} \mathbb{E}_{d_t^\pi} \mathbb{E}_\pi [A_{(1)}] \right)$$

$$= \mathbb{E}_{d_0} \mathbb{E}_\pi [A_{(0)}] + \sum_{t=0}^{T-1} \mathbb{E}_{d_t^\pi} \mathbb{E}_\pi [A_{(1)}]$$

$$3\Theta = \sum_{t=0,3,6\ldots} \mathbb{E}_{d_t^\pi} \mathbb{E}_\pi [A_{(2)}] + \left( \mathbb{E}_{d_0} \mathbb{E}_\pi [A_{(0)}] + \sum_{t=1,4\ldots} \mathbb{E}_{d_t^\pi} \mathbb{E}_\pi [A_{(2)}] \right)$$

$$+ \left( \mathbb{E}_{d_0} \mathbb{E}_\pi [A_{(1)}] + \sum_{t=2,5\ldots} \mathbb{E}_{d_t^\pi} \mathbb{E}_\pi [A_{(2)}] \right)$$

$$= \mathbb{E}_{d_0} \mathbb{E}_\pi [A_{(0)}] + \mathbb{E}_{d_0} \mathbb{E}_\pi [A_{(1)}] + \sum_{t=0}^{T-1} \mathbb{E}_{d_t^\pi} \mathbb{E}_\pi [A_{(2)}]$$

$$\vdots$$

Applying a $\lambda$-weighted over these terms, we then have

$$(1-\lambda)(1 + 2\lambda + 3\lambda^2 + \ldots)\Theta = T \mathbb{E}_{d^\pi} \mathbb{E}_\pi \left[ (1-\lambda) \sum_{i=0}^\infty \lambda^i A_{(i)} \right] + \lambda \sum_{i=0}^\infty \lambda^i \mathbb{E}_{d_0} \mathbb{E}_\pi [A_{(i)}]$$

Because for $\lambda < 1$, $\lambda + 2\lambda^2 + 3\lambda^3 + \cdots = \frac{\lambda}{(1-\lambda)^2}$, we have

$$(1-\lambda)(1 + 2\lambda + 3\lambda^2 + \ldots) = \frac{1-\lambda}{\lambda}(\lambda + 2\lambda^2 + 3\lambda^3 + \ldots) = \frac{1-\lambda}{\lambda} \frac{\lambda}{(1-\lambda)^2} = \frac{1}{1-\lambda}$$

The above derivation implies that

$$\Theta = V^\pi(d_0) - f(d_0) = (1-\lambda)T \mathbb{E}_{d^\pi} \mathbb{E}_\pi \left[ (1-\lambda) \sum_{i=0}^\infty \lambda^i A_{(i)} \right] + \lambda(1-\lambda) \sum_{i=0}^\infty \lambda^i \mathbb{E}_{d_0} \mathbb{E}_\pi \left[ A_{(i)} \right]$$

$\blacksquare$

## C.6 Proof of Lemma 3

**Lemma 3.** *For any $\lambda \in [0,1]$, any baseline value function $f : \mathcal{S} \to \mathbb{R}$, and any policy $\pi$, the following holds:*

$$h(\pi; \lambda) := (1-\lambda)T \mathbb{E}_{s \sim d^\mu} \left[ A_\lambda^{f,\pi}(s,\pi) \right] + \lambda \mathbb{E}_{s \sim d_0} \left[ A_\lambda^{f,\pi}(s,\pi) \right] \tag{15}$$

$$\nabla h(\pi; \lambda)|_{\mu=\pi} = T \mathbb{E}_{s \sim d^\pi} \mathbb{E}_{a \sim \pi|s} [\nabla \log \pi(a|s) A_\lambda^{f,\pi}(s,a)], \tag{16}$$

*where $d^\mu$ denotes the average state distribution of a policy $\mu$.*

*Proof.* We first show the gradient expression in the second term in $h(\pi; \lambda)$.

**Lemma 6.**

$$\nabla \mathbb{E}_{s \sim d_0} \left[ A_\lambda^{f,\pi}(s,\pi) \right] = \sum_{t=0}^{T-1} \lambda^t \mathbb{E}_{s_t \sim d_t^\pi} \mathbb{E}_{a_t \sim \pi|s_t} \left[ \nabla \log \pi(a_t|s_t) A_\lambda^{f,\pi}(s_t, a_t) \right]$$

*Proof of Lemma 6.* Define $Q_{(i-t)}^{f,\pi}(s_t, a_t) := \mathbb{E}_{\rho^\pi | s_t, a_t} \left[ \sum_{\tau=0}^{i-t} r(s_{t+\tau}, a_{t+\tau}) + f(s_{i+1}) \right]$. By using the definition of $i$-step advantage function $A_{(i)}^{f,\pi}$, we can first rewrite the desired derivative as

$$\nabla \mathbb{E}_{s \sim d_0} \left[ A_{(i)}^{f,\pi}(s, \pi) \right] = \nabla \mathbb{E}_{s \sim d_0} \mathbb{E}_{\rho^\pi | s_0} \left[ r(s_0, a_0) + r(s_1, a_1) + \cdots + r(s_i, a_i) + f(s_{i+1}) \right] - f(s_t)$$

$$= \sum_{t=0}^{i} \mathbb{E}_{s_t \sim d_t^\pi} \mathbb{E}_{a_t \sim \pi | s_t} \left[ \nabla \log \pi(a_t | s_t) Q_{(i-t)}^{f,\pi}(s_t, a_t) \right]$$

$$= \sum_{t=0}^{i} \mathbb{E}_{s_t \sim d_t^\pi} \mathbb{E}_{a_t \sim \pi | s_t} \left[ \nabla \log \pi(a_t | s_t) A_{(i-t)}^{f,\pi}(s_t, a_t) \right]$$

where in the last equality we use the fact $\nabla \mathbb{E}_{a \sim \pi | s}[f(s)] = 0$ for any $f : \mathcal{S} \to \mathbb{R}$. Therefore, we can write the $\lambda$-weighted version as follows:

$$\nabla \mathbb{E}_{s \sim d_0} \left[ A_\lambda^f(s, \pi) \right] = (1 - \lambda) \sum_{i=0}^{\infty} \nabla \mathbb{E}_{s \sim d_0} \left[ \lambda^i A_{(i)}^{f,\pi}(s, a) \right]$$

$$= (1 - \lambda) \sum_{i=0}^{\infty} \sum_{t=0}^{i} \lambda^i \mathbb{E}_{s_t \sim d_t^\pi} \mathbb{E}_{a_t \sim \pi | s_t} \left[ \nabla \log \pi(a_t | s_t) A_{(i-t)}^{f,\pi}(s_t, a_t) \right]$$

$$= (1 - \lambda) \sum_{t=0}^{T-1} \sum_{i=t}^{\infty} \lambda^i \mathbb{E}_{s_t \sim d_t^\pi} \mathbb{E}_{a_t \sim \pi | s_t} \left[ \nabla \log \pi(a_t | s_t) A_{(i-t)}^{f,\pi}(s_t, a_t) \right]$$

$$= (1 - \lambda) \sum_{t=0}^{T-1} \lambda^t \sum_{j=0}^{\infty} \lambda^j \mathbb{E}_{s_t \sim d_t^\pi} \mathbb{E}_{a_t \sim \pi | s_t} \left[ \nabla \log \pi(a_t | s_t) A_{(j)}^{f,\pi}(s_t, a_t) \right]$$

$$= \sum_{t=0}^{T-1} \lambda^t \mathbb{E}_{s_t \sim d_t^\pi} \mathbb{E}_{a_t \sim \pi | s_t} \left[ \nabla \log \pi(a_t | s_t) A_\lambda^{f,\pi}(s_t, a_t) \right]$$

∎

With this intermediate result, we can further derive the gradient expression in the first term in $\nabla h(\pi; \lambda)$ when $\mu = \pi$:

$$T \mathbb{E}_{s \sim d^\pi} \left[ \nabla A_\lambda^{f,\pi}(s, \pi) \right] = \sum_{t=0}^{T-1} \mathbb{E}_{s \sim d_t^\pi} \nabla \left[ A_\lambda^f(s, \pi) \right]$$

$$= \sum_{t=0}^{T-1} \sum_{\tau=t}^{T-1} \lambda^{\tau-t} \mathbb{E}_{s_\tau \sim d_\tau^\pi} \mathbb{E}_{a_\tau \sim \pi | s_\tau} \left[ \nabla \log \pi(a_\tau | s_\tau) A_\lambda^{f,\pi}(s_\tau, a_\tau) \right]$$

$$= \sum_{\tau=0}^{T-1} \mathbb{E}_{s_\tau \sim d_\tau^\pi} \mathbb{E}_{a_\tau \sim \pi | s_\tau} \left[ \nabla \log \pi(a_\tau | s_\tau) A_\lambda^{f,\pi}(s_\tau, a_\tau) \right] \left( \sum_{t=0}^{\tau} \lambda^{\tau-t} \right)$$

$$= \sum_{t=0}^{T-1} \frac{1 - \lambda^{t+1}}{1 - \lambda} \mathbb{E}_{s_t \sim d_t^\pi} \mathbb{E}_{a_t \sim \pi | s_t} \left[ \nabla \log \pi(a_t | s_t) A_\lambda^{f,\pi}(s_t, a_t) \right]$$

Finally, combining the two equalities, we arrive at a very clean expression:

$$\nabla h(\pi; \lambda)|_{\mu=\pi} = (1 - \lambda) T \mathbb{E}_{s \sim d^\pi} \left[ \nabla A_\lambda^f(s, \pi) \right] + \lambda \mathbb{E}_{s \sim d_0} \left[ \nabla A_\lambda^f(s, \pi) \right]$$

$$= \sum_{t=0}^{T-1} \mathbb{E}_{s_t \sim d_t^\pi} \mathbb{E}_{a_t \sim \pi | s_t} \left[ \nabla \log \pi(a_t | s_t) A_\lambda^{f,\pi}(s_t, a_t) \right] \left( (1 - \lambda) \frac{1 - \lambda^{t+1}}{1 - \lambda} + \lambda \cdot \lambda^t \right)$$

$$= \sum_{t=0}^{T-1} \mathbb{E}_{s_t \sim d_t^\pi} \mathbb{E}_{a_t \sim \pi | s_t} \left[ \nabla \log \pi(a_t | s_t) A_\lambda^{f,\pi}(s_t, a_t) \right]$$

∎

## C.7 Proof of Lemma 4

**Lemma 4.** *Define* $\widehat{A}(s,a) := r(s,a) + \mathbb{E}_{s'|s,a}[\widehat{f}^{\max}(s')] - \widehat{f}^{\max}(s)$. *It holds that for all* $\lambda \in [0,1]$,

$$\widehat{A}_\lambda^\pi(s_t, a_t) = \mathbb{E}_{\xi_t \sim \rho^\pi | s_t} \left[ \sum_{\tau=t}^{T-1} \lambda^{\tau-t} \widehat{A}(a_\tau, s_\tau) \right] \tag{18}$$

*Proof.* This equality can be derived as follows:

$$\widehat{A}_\lambda^\pi(s_t, a_t)$$

$$= (1-\lambda)\mathbb{E}_{\xi_t \sim \rho^\pi | s_t} \left[ \sum_{i=0}^{\infty} \lambda^i \left( \sum_{\tau=t}^{t+i} r(s_\tau, a_\tau) + \widehat{f}^{\max}(s_{t+i+1}) \right) \right] - \widehat{f}^{\max}(s_t)$$

$$= \mathbb{E}_{\xi_t \sim \rho^\pi | s_t} \left[ \sum_{\tau=t}^{T-1} r(s_\tau, a_\tau) \left( (1-\lambda) \sum_{i=\tau-t}^{\infty} \lambda^i \right) + (1-\lambda) \sum_{i=0}^{\infty} \lambda^i \widehat{f}^{\max}(s_{t+i+1}) \right] - \widehat{f}^{\max}(s_t)$$

$$= \mathbb{E}_{\xi_t \sim \rho^\pi | s_t} \left[ \sum_{\tau=t}^{T-1} \lambda^{\tau-t} r(s_\tau, a_\tau) + (1-\lambda) \sum_{\tau=t}^{T-1} \lambda^{\tau-t} \widehat{f}^{\max}(s_{\tau+1}) \right] - \widehat{f}^{\max}(s_t)$$

$$= \mathbb{E}_{\xi_t \sim \rho^\pi | s_t} \left[ \sum_{\tau=t}^{T-1} \lambda^{\tau-t} r(s_\tau, a_\tau) + \sum_{\tau=t}^{T-1} \lambda^{\tau-t} \widehat{f}^{\max}(s_{\tau+1}) - \sum_{\tau=t+1}^{T-1} \lambda^{\tau-t} \widehat{f}^{\max}(s_\tau) \right] - \widehat{f}^{\max}(s_t)$$

$$= \mathbb{E}_{\xi_t \sim \rho^\pi | s_t} \left[ \sum_{\tau=t}^{T-1} \lambda^{\tau-t} \left( r(s_\tau, a_\tau) + \widehat{f}^{\max}(s_{\tau+1}) - \widehat{f}^{\max}(s_\tau) \right) \right]$$

$$= \mathbb{E}_{\xi_t \sim \rho^\pi | s_t} \left[ \sum_{\tau=t}^{T-1} \lambda^{\tau-t} \widehat{A}(s_\tau, a_\tau) \right]$$

∎

## C.8 Proof of Theorem 2

**Theorem 2.** *Performing no-regret online learning w.r.t.* (12) *has the guarantee in Theorem 1, except now* $\epsilon_N(\Pi)$ *can be negative when* $\lambda > 0$.

*Proof.* The proof Theorem 2 is based on the non-trivial technical lemma of this general $\lambda$-weighted advantage, which we recall below.

**Lemma 2.** *For any policy* $\pi$, *any* $\lambda \in [0,1]$, *and any baseline value function* $f : \mathcal{S} \to \mathbb{R}$,

$$V^\pi(d_0) - f(d_0) = (1-\lambda)T\mathbb{E}_{s \sim d^\pi} \left[ A_\lambda^{f,\pi}(s, \pi) \right] + \lambda\mathbb{E}_{s \sim d_0} \left[ A_\lambda^{f,\pi}(s, \pi) \right], \tag{14}$$

*where* $A_\lambda^{f,\pi}$ *is defined like* $A_\lambda^{\max,\pi}$ *but with a general* $f$ *instead of* $f^{\max}$.

To prove the theroem, we can then write down an equality like (8) by the equality $V^{\pi_n}(d_0) - f^{\max}(d_0) = \ell_n(\pi_n; \lambda)$ we just obtained:

$$\frac{1}{N} \sum_{n \in [N]} V^{\pi_n}(d_0) = f^{\max}(d_0) + \Delta_N - \epsilon_N(\Pi) - \frac{\text{Regret}_N}{N}$$

where $\text{Regret}_N$, $\Delta_N$, and $\epsilon_N(\Pi)$ are now defined with respect to the $\lambda$-weighted online loss $\ell_n(\pi; \lambda)$. Therefore, running a no-regret algorithm with respect to the approximate gradient (17) of this online loss function $\ell_n(\pi; \lambda)$ would imply a similar performance guarantee shown in Theorem 1 (see the proof Theorem 1).

Finally, to justify the use of (17), what remains to be shown is that the term $\Delta_N - \epsilon_N(\Pi)$ behaves similarly as before. First we notice that, because $\pi^{\max}$ may not be the best policy for the multi-step advantage in the online loss $\ell_n(\pi; \lambda)$, $\epsilon_N(\Pi)$ now be negative (which is in our favor). Next, we show that $\Delta_N \geq 0$ is true by Proposition 2 (whose proof is given below).

**Proposition 2.** *It holds* $-\ell_n(\pi^{\max}; \lambda) \geq 0$.

These results conclude the proof of Theorem 2.

∎

### C.8.1 Proof of Proposition 2

*Proof.* We first prove a helpful lemma.

**Lemma 7.** *For* $\pi^{\max}$, *it holds that* $A^{\max}_{(i)}(s, \pi^{\max}) \geq 0$.

*Proof of Lemma 7.* Without loss of generality, take $s = s_0$. First we arrange

$$A^{\max}_{(i)}(s_0, \pi^{\max}) = \mathbb{E}_{\rho^{\pi^{\max}}|s_0} \left[ r(s_0, a_0) + r(s_1, a_1) + \cdots + r(s_i, a_i) + V^{k_{s_{i+1}}}(s_{i+1}) \right] - V^{k_{s_0}}(s_0)$$

$$= \mathbb{E}_{\rho^{\pi^{\max}}|s_0} \left[ r(s_0, a_0) + r(s_1, a_1) + \cdots + Q^{\max}(s_i, \pi^{\max}) \right] - V^{k_{s_0}}(s_0)$$

where we have the inequality

$$Q^{\max}(s_i, \pi^{\max}) := \mathbb{E}_{a_i \sim \pi^{\max}|s_i}[r(s_i, a_i) + \mathbb{E}_{s_{i+1} \sim \mathcal{P}|s_i, a_i}[V^{k_{s_{i+1}}}(s_{i+1})]]$$

$$\geq \mathbb{E}_{a_i \sim \pi^{k_{s_i}}|s_i}[r(s_i, a_i) + \mathbb{E}_{s_{i+1} \sim \mathcal{P}|s_i, a_i}[V^{k_{s_{i+1}}}(s_{i+1})]]$$

$$\geq \mathbb{E}_{a_i \sim \pi^{k_{s_i}}|s_i}[r(s_i, a_i) + \mathbb{E}_{s_{i+1} \sim \mathcal{P}|s_i, a_i}[V^{k_{s_i}}(s_{i+1})]]$$

$$= V^{k_{s_i}}(s_i)$$

By applying this inequality recursively, we get

$$A^{\max}_{(i)}(s_0, \pi^{\max}) \geq V^{k_{s_0}}(s_0) - V^{k_{s_0}}(s_0) \geq 0$$

∎

The lemma above implies $A^{\max}_{\lambda}(s, \pi^{\max}) \geq 0$ for $\lambda \geq 0$ and therefore we have

$$-l_n(\pi^{\max}; \lambda) = (1 - \lambda)T\mathbb{E}_{s \sim d^{\pi_n}}\left[A^{\max}_{\lambda}(s, \pi^{\max})\right] + \lambda\mathbb{E}_{s \sim d_0}\left[A^{\max}_{\lambda}(s, \pi^{\max})\right] \geq 0$$

∎

## D  Experiment Details and Additional Results

In this section we describe the details of MAMBA and additional experimental results.

### D.1  Implementation Details of MAMBA

We provide the details of MAMBA in Algorithm 1 as Algorithm 2, which closely follows Algorithm 1 with a few minor, practical modifications which we describe below:

- The UpdateInputWhitening function keeps a moving average of the first and the second moment of the states it has seen, which is used to provide whitened states (by subtracting the estimated mean and dividing by the estimated standard deviation) as the input to the neural network policy.

- In Algorithm 2, $t_e = \text{SampleSwitchTime}(t_{\text{avg}})$ samples $t_e$ based on a geometric distribution whose mean is $t_{\text{avg}}$, because in the learner might not always be able to finish the full $T$ steps. The trajectory data are therefore suitably weighted by an importance weight $\frac{1}{Tp(t_e)}$ to correct for this change from using the uniform distribution in sampling for $t_e$.

Apart from these two changes, Algorithm 2 follows the pseudo-code in Algorithm 1.

We provide additional details of the experimental setups below.

---

**Algorithm 2** Implementation details of MAMBA for IL with multiple oracles

---

**Input:** Initial learner policy $\pi_1$, oracle polices $\{\pi^k\}_{k\in[K]}$, function approximators $\{\widehat{V}^k\}_{k\in[K]}$.
**Output:** The best policy in $\{\pi_1, \ldots, \pi_N\}$.
1: For each $k \in [K]$, collect data $\mathcal{D}^k$ by rolling out $\pi^k$ for the full problem horizon.
2: Update value models $\widehat{V}^k = \text{MonteCarloRegression}(\mathcal{D}^k)$ for $k \in [K]$.
3: $\pi_1 = \text{UpdateInputWhitening}(\pi_1, \bigcup \mathcal{D}^k)$
4: **for** $n = 1 \ldots N - 1$ **do**
5:     Sample a trajectory using $\pi_n$ to collect data $\mathcal{D}'_n$.
6:     Let $t_e = \text{SampleSwitchTime}(t_{\text{avg}}) \in [T-1]$, where $t_{\text{avg}}$ is the average trajectory length of $\mathcal{D}'_{1:n-1}$ (for the first iteration set $t_{\text{avg}} = 0$). Sample a RIRO trajectory using $\pi_n$ and then $\pi^k$ after $t \geq t_e$ to collect data $\mathcal{D}_{\text{RIRO}}$, where $k$ is uniformly sampled in $[K]$. If the sampled trajectory in $\mathcal{D}_{\text{RIRO}}$ is longer than $t_e$, aggregate the trajectory after $t_e$ in $\mathcal{D}_{\text{RIRO}}$ into $\mathcal{D}^k$ with importance weight $\frac{1}{Tp(t_e)}$. Otherwise, aggregate $\mathcal{D}_{\text{RIRO}}$ into $\mathcal{D}'_n$.
7:     Update value model $\widehat{V}^k = \text{MonteCarloRegression}(\mathcal{D}^k)$.
8:     Let $\pi'_n = \text{UpdateInputWhitening}(\pi_n, \mathcal{D}'_n)$ and compute the sampled gradient based on $\mathcal{D}'_n$ with one-step importance sampling as

$$g_n = -\sum_{t=0}^{T-1} \nabla \log \pi'_n(a_t|s_t) \frac{\pi'_n(a_t|s_t)}{\pi_n(a_t|s_t)} \left( \sum_{\tau=t}^{T-1} \lambda^{\tau-t} \left( r(s_\tau, a_\tau) + \widehat{f}^{\max}(s_{\tau+1}) - \widehat{f}^{\max}(s_\tau) \right) \right)$$

    where the recursion starts with $\widehat{V}^k(s_T) = 0$ and $\widehat{f}^{\max}(s) = \max_{k\in[K]} \widehat{V}^k(s)$.
9:     Update the policy $\pi_{n+1} = \text{MirrorDescent}(\pi'_n, g_n)$.
10: **end for**

---

- Time is appended as a feature in addition to the raw state, i.e. $s = (t, \bar{s})$.

- The policy is a Gaussian with mean modeled by a $(128, 128)$ FCNN (fully connected neural network), and the standard deviation is diagonal, learnable and independent of the state. The value function is estimated by a $(256, 256)$ FCNN. In these FCNNs, the activation functions are $tanh$ except the last layer is linear. The policy and the value networks are parameterized independently.

- MonteCarloRegression performs weighted least-squared regression by first whitening the input and then performing 100 (CartPole, DoubleInvertedPendulum) or 800 (HalfCheetah, Ant) steps of ADAM with a batchsize of 128 samples and step size 0.001. The target is constructed by Monte-Carlo estimate and one-step importance sampling, if necessary.

- MirrorDescent is set to be either ADAM [37] or Natural Gradient Descent (NGD) [38]. We adopt the default hyperparamters of ADAM ($\beta_1 = 0.9$ and $\beta_2 = 0.99$) and a stepsize 0.001. For NGD, we adopt the ADAM-style adaptive implementation described in [43, Appendix C.1.4] using $\beta_2 = 0.99$ and a stepsize of 0.1.

- $\mathcal{D}^k$ is limited to data from the past 100 (CartPole, DoubleInvertedPendulum) or 2 (HalfCheetah, Ant) iterations. Policy gradient always keeps 2 iterations of data.

- All MAMBA, PG-GAE, and AggreVaTeD follow the protocol in Algorithm 2. In the pre-training phase (lines 1-3), each oracle policy (or the learner policy for PG-GAE) would perform 16 (CartPole, DoubleInvertedPendulum) or 512 (HalfCheetah, Ant) rollouts to collect the initial batch of data to train its value function estimator. For HalfCheetah and Ant, these data are also used to pretrain the learner policies by behavior cloning[7] In every iteration, each algorithm would perform 8 (CartPole, DoubleInvertedPendulum) or 256 (HalfCheetah, Ant) rollouts: For MAMBA and AggreVaTeD, half rollouts are used to collect data for updating the value functions (line 6) and half rollouts (line 5) are for computing the gradient. For PG-GAE, all rollouts are performed by the learner; they are just used to compute the gradient and then update the value function (so there is no correlation).

- Additional 8 (CartPole, DoubleInvertedPendulum) or 256 (HalfCheetah, Ant) rollouts are performed to evaluate the policy's performance, which generate the plots presented in the paper.

- All environments have continuous states and actions. CartPole and DoubleInvertedPendulum have a problem horizon of 1000 steps. HalfCheetah and Ant have a problem horizon of 500 steps. For CartPole, the dimensions of the state and the action spaces are 4 and 1, respectively. For DoubleInvertedPendulum, the dimensions of the state and the action spaces are 8 and 1, respectively. For HalfCheetah, the dimensions of the state and the action spaces 17 and 6, respectively. Finally, for Ant, the dimensions of the state and the action spaces 111 and 8, respectively.

**Computing Infrastructure** The CartPole and DoubleInvertedPendulum experiments were conducted using Azure Standard F64s_v2, which was based on the Intel Xeon Platinum 8168 processor with 64 cores (base clock 3.4 GHz; turbo clock 3.7 GHz) and 128 GB memory. The HalfCheetah and Ant experiments were conducted using Azure Standard HB120rs_v2, which was based on the AMD EPYC 7002-series processor with 120 cores (base clock 2.45 GHz; turbo clock 3.1 GHz) and 480 GB memory. No GPU was used. The operating system was Ubuntu 18.04.4 LTS. The prototype codes were based on Python and Tensorflow 2. Using a single process in the setup above, a single seed of the CartPole experiment (100 iterations) took about 27 minutes to 45 minutes to finish, whereas a single seed of the DoubleInvertedPendulum experiment (200 iterations) took about 110 minutes to 125 minutes to finish. For HalfCheetah, 8 cores were used for each seed (1000 iterations) and the experiments took about 19.5 to 30.7 hours. For Ant, 8 cores were used for each seed (400 iterations) and the experiments took about 8 to 12 hours.

**Hyperparameter Selection** For CartPole and DoubleInvertedPendulum, we only performed a rough search of the step sizes of ADAM (0.01 vs 0.001) and Natural Gradient Descent (0.1 vs 0.01). We conducted experiments with different $\lambda$ and number of oracles in order to study their effects on MAMBA. The main paper presents the results of the pilot study: we first investigated the effect of $\lambda$ by comparing MAMBA with AggreVaTeD and concluded with a choice of $\lambda = 0.9$; using this $\lambda$ value, we then studied the effects of the number of oracles. For completeness, we present and discuss results of all the hyperparamters for CartPole and DoubleInvertedPendulum below. For HalfCheetah, we did a hyperparamter search over the size of replay buffer and the optimization steps for value function fitting such that AggreVaTeD can achieve the oracle-level performance. Once that is chose, we apply the same hyperparamters to other algorithms. For Ant, we simply apply the same hyperparamters used in HalfCheetah.

**Oracle Performance** For Cartpole and DoubleInvertedPendulum, please see the detailed discussion in the next section. For HalfCheetah, the four oracles used in the experiments have scores of 1725.80, 1530.85, 1447.84, and 1395.36, respectively. For Ant, the four oracles used in the experiments have scores of 1050.57, 1038.03, 883.18, 775.70, respectively.

## D.2 Additional Experimental Results of Hyperparamter Effects

In this section, we include in Figs. 6 to 9 additional experimental results of CartPole and DoubleInvertedPendulum environments. The purpose of these extra results is to provide a more comprehensive picture of the properties of MAMBA under various hyperparamter settings,.

**Setup** For each of the environments (CartPole and DoubleInvertedPendulum), we conduct experiments with Bad Oracles (Figs. 6 and 8) and Mediocre Oracles (Figs. 7 and 9), where the results of the Bad Oracles are the ones presented in the main paper. In each experiment, we run MAMBA with $\lambda = 0, 0.1, 0.5$ and $0.9$, and with the number of oracles varying between $1, 2, 4$ and $8$. In addition, we run AggreVaTeD with each of the oracles (whereas the main paper only presents the results of the oracles with the highest return). Finally, for baselines, we include the learning curve of PG-GAE as well as the return of each oracle. Recall that the oracles are indexed in a descending order of their returns, which are estimated by performing 8 rollouts.

### D.2.1 Brittleness of AggreVaTeD

First, the experiments of AggreVaTeD highlight that performing IL via policy improvement[8] from the best oracle (in terms of the return) does not always lead to the best learner policy, as constantly

(a) $\lambda = 0$     (b) $\lambda = 0.1$

(c) $\lambda = 0.5$     (d) $\lambda = 0.9$

(e) AggreVaTeD     (f) Oracle Performance

|         | Return |
|---------|--------|
| oracle0 | 89.04  |
| oracle1 | 61.73  |
| oracle2 | 34.78  |
| oracle3 | 26.34  |
| oracle4 | 18.19  |
| oracle5 | 16.36  |
| oracle6 | 10.38  |
| oracle7 | 9.78   |

Figure 6: Performance of the best policies in CarlPole with Bad Oracles. (a)-(d) MAMBA with $\lambda = 0, 0.1, 0.5, 0.9$ (e) AggreVaTeD with different oracles. (f) The return of each oracle policy. A curve shows an algorithm's median performance across 8 random seeds. The center and right figures use the same line colors for all methods. The shaded area shows 25th and 75th percentiles.

shown in Figs. 6 to 9. In general there is no upper bound on the amount of performance change that policy improvement can make, because the policy improvement operator is myopic, only looking at one step ahead. As a result, running AggreVaTeD with the best oracle does not always give the best performance that can be made with an oracle chosen in the hindsight. Another factor to the differences between the best foresight and hindsight oracles is that the return of each oracle is only estimated by 8 rollouts here.

Our experimental results show that such sensitivity is reduced in MAMBA: even in the single-oracle setting, using a $\lambda > 0$ in MAMBA generally leads to more robust performance than AggreVaTeD using the same, best oracle, which is namely MAMBA with $\lambda = 0$. We should remark that this robustness is not fully due to the bias-variance trade-off property [17], but also by large attributed to the incorporation of the multi-step information in online loss in (12) (cf. Theorem 2). By using $\lambda > 0$, MAMBA can start to see beyond one-step improvement and becomes less dependent on the oracle quality. In the experiments, we observe by picking a large enough $\lambda$, MAMBA with a single oracle usually gives comparable if not better performance than AggreVaTeD with the best policy chosen in the *hindsight*.

Figure 7: Performance of the best policies in CarlPole with Mediocre Oracles. (a)-(d) MAMBA with $\lambda = 0, 0.1, 0.5, 0.9$ (e) AggreVaTeD with different oracles. (f) The return of each oracle policy. A curve shows an algorithm's median performance across 8 random seeds. The center and right figures use the same line colors for all methods. The shaded area shows 25th and 75th percentiles.

### D.2.2 Effects of $\lambda$-weighting

Beyond the single-oracle scenario discussed above, we see consistently in Figs. 6 to 9 that using a non-zero $\lambda$ improves the performance of MAMBA. The importance of $\lambda$ is noticeable particularly in setups with Bad Oracles, as well as in the experiments with the higher-dimensional environment DoubleInvertedPendulum. Generally, when the oracles are bad (as in Figs. 6 and 8), using a larger $\lambda$ provides the learner a chance to outperform the suboptimal oracles as suggested by Theorem 2, because the online loss function in (12) starts to use multi-step information. On the other hand, when the top oracles' performance is better and the state space is not high-dimensional, as in CartPole with Mediocre Oracles in Fig. 7, the effects of $\lambda$ is less prominent. The usage of $\lambda > 0$ also helps reduce the dependency on function approximation error, which is a known GAE property [17], as we see in the experiments with DoubleInvertedPendulum in Figs. 8 and 9.

### D.2.3 Effects of multiple oracles

Using more than one oracles generally lead to better performance across Figs. 6 to 9. In view of Theorem 2, using more oracles can improve the quality of the baseline value function, though at the cost of having a higher bias in the function approximators (because more approximators need to be learned). We see that the benefit of using more oracles particularly shows up when higher values of $\lambda$ are used; the change is smaller in the single-oracle settings.

(a) $\lambda = 0$

(b) $\lambda = 0.1$

(c) $\lambda = 0.5$

(d) $\lambda = 0.9$

(e) AggreVaTeD

| | Return |
|---|---|
| oracle0 | 4244.16 |
| oracle1 | 3408.81 |
| oracle2 | 2775.02 |
| oracle3 | 2440.19 |
| oracle4 | 2329.90 |
| oracle5 | 2177.31 |
| oracle6 | 859.82 |
| oracle7 | 712.61 |

(f) Oracle Performance

Figure 8: Performance of the best policies in DoubleInvertedPendulum with Bad Oracles. (a)-(d) MAMBA with $\lambda = 0, 0.1, 0.5, 0.9$ (e) AggreVaTeD with different oracles. (f) The return of each oracle policy. A curve shows an algorithm's median performance across 8 random seeds. The center and right figures use the same line colors for all methods. The shaded area shows 25th and 75th percentiles.

However, in the settings with Mediocre Oracles in Figs. 7 and 9, increasing the number of oracles beyond a certain threshold degrades the performance of MAMBA. Since a fixed number of rollouts are performed in each iteration, having more oracles implies that the learner would need to spend more iterations to learn the oracle value functions to a fixed accuracy. In turn, this extra exploration reflects as slower policy improvement. Especially, because using more oracles here means including strictly weaker oracles, this phenomenon is visible, e.g., in Fig. 9.

### D.3  Additional Experimental Results of Oracle Ordering

In all the previous experiments, we order the oracles based on the their performance in terms of their return. However, these return estimates are only empirical and do not always correspond to the true ordering of the oracles, as we discussed in Appendix D.2.1. To study the robustness to oracle selection, here we randomly order the oracles before presenting them to the IL algorithms (MAMBA and AggreVaTeD) and repeat the controlled experiment of testing the effects of $\lambda$-weighting and the number of oracles in Fig. 3. The results of random oracle ordering are presented in Fig. 10; because of this extra randomness we inject in oracle ordering, we use more seeds in these experiments. First, we see in Fig. 10a using the random ordering degrades of the performance of the single-oracle setup. This is reasonable because there is a high chance of selecting an extremely bad oracle (see Fig. 6 for the oracle quality). Nonetheless, the usage of $\lambda > 0$ still improves the performance: the learning is

| | Return |
|---|---|
| oracle0 | 9453.84 |
| oracle1 | 9079.43 |
| oracle2 | 6193.54 |
| oracle3 | 5791.93 |
| oracle4 | 4227.62 |
| oracle5 | 4206.01 |
| oracle6 | 1535.97 |
| oracle7 | 1480.38 |

(a) $\lambda = 0$    (b) $\lambda = 0.1$    (c) $\lambda = 0.5$    (d) $\lambda = 0.9$    (e) AggreVaTeD    (f) Oracle Performance

Figure 9: Performance of the best policies in DoubleInvertedPendulum with Mediocre Oracles. (a)-(d) MAMBA with $\lambda = 0, 0.1, 0.5, 0.9$ (e) AggreVaTeD with different oracles. (f) The return of each oracle policy. A curve shows an algorithm's median performance across 8 random seeds. The center and right figures use the same line colors for all methods. The shaded area shows 25th and 75th percentiles.

faster and converges to higher performance, though it is still slower than PG-GAE because of the extremely bad oracles.

But interestingly once we start to use *multiple* oracles in Fig. 10b, MAMBA starts to significantly outperform AggreVaTeD and PG-GAE. By using more than one oracle, there is a higher chance of selecting one reasonable oracle in the set filled with bad candidates. In addition, the diversity of oracle properties also help strengthen the baseline value function (cf. Theorem 2). Thus, overall we observe that MAMBA with $\lambda > 0$ and multiple oracles yields the most robust performance.

(a) Effects of $\lambda$-weighting

(b) Effects of number of oracles

Figure 10: Performance of the best policies with *random orderings of oracles* in CartPole with Bad Oracles. (a) shows the single-oracle setup comparing MAMBA with different $\lambda$ values. (b) show MAMBA with different number of oracles ($\lambda = 0.9$). A curve of MAMBA and AggreVaTeD shows the performance across 32 random seeds. The curves of PG-GAE shows the performance across 8 random seeds. The shaded area shows 25th and 75th percentiles.

## Footnotes

[7]We found that without behavior cloning AggreVaTeD would not learn to attain the oracle's performance, potentially due to high dimensionality. Therefore, we chose to perform behavior cloning with the best oralce policy as the initialization for all IL algorithms in comparison here.

[8] AggreVaTeD is an approximate policy improvement algorithm [14].