[Reviews · NeurIPS 2020]

Review 1

Summary and Contributions: This paper considers the problem of learning from a number of sub-optimal demonstration policies. It proposes an imitation learning algorithm, MAMBA, which at any state, “follows” the demonstration policy with the highest value at the state.

Strengths: > The considered problem, studying imitation learning with multiple expert policies, is highly relevant, and the proposed solution, while conceptually simple, will be of value to the NeurIPS community > The paper is well-written, and the main manuscript strikes a nice balance between formal rigor and relevant intuitions.

Weaknesses: > The empirical evaluation is on a particularly weak set of domains that don’t convincingly demonstrate the effectiveness of the algorithm -- I would like to see evaluation on more challenging domains.

Correctness: I did not examine the proofs in C.5, but the rest of the proofs seem correct. The empirical methodology also seems correct to me.

Clarity: The paper is written well and easy to follow.

Relation to Prior Work: The paper positions itself well to the best of my knowledge.

Reproducibility: Yes

Additional Feedback: Update: I've read the rebuttal and other reviews. The rebuttal did answer some of my general questions, especially around estimating value functions, and I still stand by my assessment to accept the paper. > Perhaps only a minor nitpick in notation, but I found the use of the word “expert” a bit dissonant -- the paper seems to only consider non-optimal policies, and especially in the empirical evaluation the “expert” policies are not close in any sense to “expert”. > L115 says you do not assume that expert actions are observed: are value functions for the experts estimated using MC in this setting? > L141: Doesn’t value function estimation (for the experts) incur complexity dependent on the horizon; if so, how can this algorithm find a policy as good as the expert policies in complexity independent of the horizon”? > Off the previous point, I’m a little concerned by the need for this algorithm to actually estimate advantages and values, as compared to for example \pi^\dot (in Eq3) which only requires learning at each state *which* policy has the highest value, and not needing to compute the value itself. Perhaps a meta-point, I understand from a practical perspective, imitation learning algorithms like DaGGER / BC are perhaps favored since the algorithms do not need to incur the challenges in value function estimation. > Figure 1 caption: polices -> policies > The empirical evaluation is quite thorough and detailed well in the appendix


Review 2

Summary and Contributions: This paper tackles the problem of imitation learning from multiple experts (in the context of RL), where the experts are not necessarily optimal, may have varying performance, or may be better than one-another in different regions of the state-space. More specifically, they tackle the problem setting in which experts are available online, and the MDP's rewards are available, as opposed an imitation learning framework in which experts generate a collection of demonstrations ahead of time and the MDP rewards are not available. The core contribution of the author's work is extending prior work on imitation learning from an online expert with rewards (Aggrevate/D) to the setting where multiple, not totally optimal, experts are available. The technical contributions seem sound.

Strengths: - Given the density of content, the paper is well-written. What I find quite good about the presentation is the pedagogical nature of the writing, starting with making the most set of assumptions, and gradually removing them, showing the modification necessary for removing each assumption. Significance: High - First a sound algorithmic choice is presented and analyzed (pi^dot, section 3.1), and then the potential shortcomings of this approach are pointed out, which motivates the development of the main algorithm (pi^max). While the modification from pi^dot to pi^max in itself appears straightforward, I believe the care taken to demonstrate why pi^max is a better choice is valuable. Additionally, it is nice that pi^max does not need to observe the expert actions (although if I am correct Aggrevate did not either?) Significance: Medium/High - I personally had not seen much work on learning from multiple experts, so it is a valuable contribution to add to an under-explored domain. Also, according to the related works section in the appendix, prior approaches to multiple experts do not have theoretical analyses and performance guarantees.

Weaknesses: Highest priority comments are the P0 comments listed below. P0: - I think you should clarify what you mean by "experts". You are allowing the definition of experts to include sub-optimal policies, but is there an extent to which you are allowing them to be suboptimal? I feel like this needs to be clarified. If they can be any policy, then does this not fall more in the domain of off-policy/batch RL, rather than imitation learning. Some works from that literature, for example Advantage Weighted Regression by Peng et al. might have an algorithmic form that seems to have some connections to your work. Adding comments regarding connections to batch RL in the related works section might be a good idea. If there is no strict definition for sub-optimal, then maybe you should run some experiments where you also add some low-performing policies to your mix of "experts" and demonstrate how that influences your empirical results. - Missing related works section in the main text. I think the best place for this would be right before the experiments section so that exact connections to prior algorithms could be drawn, after having established your method. - Line 235, Equation 12: why is the lambda in equation 12, the same lambda in equation 13 for the GAE-style estimates? - When lambda = 1, the first part of eq 2 is 0, and the second part is 0 by eq 13. When lambda = 0, the second part of eq 12 is 0, and the first part is 0 because of eq 13. So when lambda = 0/1, the loss function in equation 12 is 0. Am I making a mistake/missing something, or is there something wrong in your equations? - Line 243: could you clarify why lambda = 1 is RL? - Line 246, equation 14: I had a quick look at the appendix (line 553) and it seems that this is true only when the derivate is evaluated at pi_n. Am I missing something? - Lines 249-252: Could you expand this paragraph and provide a bit more explanation? - Equation 12 is justified because of the form of its gradient in equation 14. As your manuscript already has a nice pedagogical writing style, I was wondering the conceptual jump from prior equations to equation 12 could be better motivated. For example, you could answer the following question: Given the discussion thus far, what motivated you/gave you the idea to write down this form of equation? - It would be good to somehow show the expert performance in Figures 1 and 2. For example, using dotted horizontal lines depicting max and min expert performances? - I don't think you described how you obtained your "experts". P1: - I think it would be valuable is you also did some experiments on slightly harder problems, such as the standard benchmark Mujoco domains: HalfCheetah, Ant, Hopper, Walker. Because then we could get a better sense of how big the value of multiple experts is in harder domains than cartpole and doublependulum. It would be particularly interesting I think if you also had experiments in these domains that used multiple experts of varying performances (e.g. checkpointing models at 1000, 2000, 3000, 4000 returns and using them as your mix of experts (these specific numbers are arbitrary)). I think this would help the adoption of your work if you can show gains with this problem setup. Other comments: - Figure 1: move closer to experiments section - Line 59: typo, is --> in - Line 246: typo, now it is now --> it is now - Line 274, footnote 5: Isn't natural gradient a second-order method?

Correctness: Claims and methods seem reasonable, although I have to say that I have not verified theorems, etc., and have some comments in the weaknesses section about some of the equations that I would like to get addressed. For empirical methodology, I have a slight concern. In the MAMBA algorithm box (Algorithm 2), there are some aspects that are influenced by the number of experts available, that may skew the results unfairly when more experts are available. For example, in line 3 UpdateInputWhitening will have access to more data when there are more experts available. Could you run some experiments, while accounting for such differences to make sure the empirical performance differences are not simply due to these.

Clarity: Yes, given the density of material, the manuscript is well-written in an informative and pedagogical manner.

Relation to Prior Work: - This needs to be brought inside the main text and not relegated to the appendix section. I think the best place for this would be right before the experiments section so that exact connections to prior algorithms could be drawn, after having established your method.

Reproducibility: Yes

Additional Feedback: With adequate response to the questions and weaknesses discussed, I am fairly inclined to increase my score as I thought the authors have written a good manuscript with nice contributions. <<< UPDATES AFTER REVEIWER DISCUSSION PERIOD >>> Even before I was inclined to give a 7 for this paper, as I thought it was a good contribution and well-written for the most part. Although I had done work in imitation learning/IRL, I did not have much knowledge/experience of imitation learning with rewards present. Before I read your paper, I had a look at Aggrevate/D, and while reading your work, I felt like I learned interesting concepts. For this reason I am increasing my score to an 8. Another reason for this increase of score is because of some clarifications in the rebuttal as well as the additional inclusion of experiments in harder domains (humanoid). I would still like to emphasize the following to the reviewers. If your paper is accepted ( hopefully it will be :) ), I **strongly** encourage you to (If accepted, I think you get an extra 9th page for your main manuscript, so there should be enough room): - Expand upon the related works section of your paper and **include it in the main manuscript** - I strongly encourage a rewriting/restructuring of section 4. Most of the work was well-written, with the exception of this part, which is your main algorithm. Some suggestions are the following: 1) Clarify the limiting cases of choice of \lambda. I see your point about \lambda = 0. But then \lambda --> 1 is still not very clear. Naively looking at equation 13, plugging 1, it look like it would be zero because of the first coefficient of (1-\lambda). I can see from your rebuttal that since it's an infinite sum, there's some limit effects happening, but for the reader it would be very useful for your to actually clarify it. 2) Think about how to rewrite lines 249-256. Sentences like "Controlling ..." in line 250 might make sense to you as the authors, but a reader who hasn't spent a lot of time doing the research for this work, it takes a bit of time and guessing to see what you mean. - Please include any experiments with harder domains in the **main manuscript** as well (and maybe compare with one state-of-the-art RL method to show the improvement in speed of learning). Your choice of humanoid domain should be a good one since it's one of the harder ones for RL, if you used HalfCheetah etc. the delta in performance when using experts might not be too high. - I still think your work could be comparable to batch RL algorithms in spirit, but in a setting where experts are available online. It's still unclear to me how sub-optimal you can let your experts be and still learn well? If I would design one single experiment to test this, I would probably go with my hardest domain, humanoid in your case, train a couple of experts up to different levels of performance (10%, 25%, 50%), and see what gains you can get. This is aligned with how batch RL experiments are being standardized as well (e.g. the D4RL dataset of Fu et al.). Potential algorithms you could compare to could be "Learning Complex Dexterous Manipulation with Deep Reinforcement Learning and Demonstrations" by Rajeswaran et al. Thank you for your submission. I recommend an accept. :)


Review 3

Summary and Contributions: This paper describes an algorithm called MAMBA to learn from and improve over multiple policies. The algorithm is theoretically justified and is validated empirically.

Strengths: The theoretical results show that the approach has a small regret bound. The empirical results show that it performs better than previous methods such as AggraVaTeD in Cart-pole and Double-Inverted Pendulum.

Weaknesses: Comparison in more domains would have strengthened the paper.

Correctness: The empirical methodology is sound. The theory looks correct, but needs more justification,

Clarity: The paper is overall well written, but some parts are unclear. More explanation of pages 6 and 7 will be useful, e.g., in the paragraph below equation 11, footnote 3 is useful but lines 223-225 are not clear to me. In page 7, it is said that the first part of theorem holds for any f. It might make sense to make it as a separate Lemma and state for any f.

Relation to Prior Work: Prior work is well-described.

Reproducibility: Yes

Additional Feedback: Page 6, lines 207-208. Why is this claim true? In particular it is not clear where o(N) is coming from. It cannot mean that there is bound to be some non-zero improvement in every step. The text needs to have more explanation connecting Algorithm 1 to the loss function (12). In particular, why the two trajectories are sampled the way they are. I have read the authors' response and am satisfied with it. I retain my scores.

[Author Response · NeurIPS 2020]

## 1 Common Questions

*"Expert" policies.* We now see how the term "expert" can be misleading. In the revision, we will replace it with "oracle" and change the paper title to "Policy Improvement via Imitation of Multiple Oracles". While we allow weaker oracle policies than is typical in imitation learning literature, they still need to have meaningful behaviors in order to provide an informative advantage function for policy update. For example, they cannot be completely uniform behavior policies as is often done in the batch RL setting.

*Extra experiments.* We plan to include more results on harder domains. In Fig. 1, we present a preliminary results of Humanoid-v2 in Mujoco, where we downloaded the pretrained policy from OpenAI baseline as the oracle. We see MAMBA significantly improves learning in this single expert case, as it uses $\lambda$ compared with AggreVaTeD and uses the oracle value compared with PG-GAE. (The initial policy for MAMBA and AggreVaTeD are pretrained with BC).

## 11 Reviewer 1

*Estimation of values (L115).* The values of the oracle policies are estimated by MC in line 4 of Algorithm 1. They are used as the target to train a neural-net $\widehat{V}^k$, which is later used in the gradient computation in line 6 for policy update.

*Value estimation complexity (L141).* Correct, it depends on the horizon, which is captured by $\beta$ and $\nu$ in Thm 1. The remark in line 141 is for the idealized case when the value functions (or rather a full MDP model) are known. Gradient estimation bias and variance can increase with horizon when learning is involved. We will further clarify this in the revision.

Figure 1: Results of 4 seeds. The values of $\lambda$ are given in the legend.

*Comparison with $\pi^{\bullet}$.* We remark that both $\pi^{\bullet}$ and $\pi^{\max}$ can be learned without explicit value estimation. For example, once we have an estimate of $\pi^{\bullet}$ for picking the best oracle in a given state, we can use it in an RIRO setting to estimate $V^{\max}(s')$ and construct $r(s,a) + \mathbb{E}[V^{\max}(s')]$, which provides enough information to learn $\pi^{\max}$ (since changing the state dependent baseline in the advantage doesn't affect the policy derived from it). However this approach suffers from high variance in our experiences; that is why we designed MAMBA based on value function estimators. We recall also that $\pi^{\max}$ corresponds to one-step policy improvement while $\pi^{\bullet}$ corresponds to behavior cloning in the idealized case of a known value function with only one expert, hence we end up not considering $\pi^{\bullet}$.

*Comparison with BC/DAgger.* While MAMBA does require value estimation, this lets MAMBA 1) combine multiple oracle policies' strengths and 2) further improve upon their performance. Without the value information, this is impossible in general, because there is no signal to distinguish the quality or preference of different oracle policies.

## 30 Reviewer 2

*Related work.* Thanks for extra pointers! We'll move more rel. work discussion to the main text and include them there.

*Properties of $\lambda$.* Note that Eq (13) provides the definition of $A_{\lambda}^{max,\pi}$ used in Eq (12). For (13), when $\lambda = 1$, because the sum in Eq (13) is *infinite* and the problem horizon $T$ is *finite*, we have $A_{\lambda}^{max,\pi}(s_t, a_t) = \mathbb{E}_{\rho^{\pi}}[\sum_{\tau=t}^{T-1} r(s_{\tau}, a_{\tau})] - f^{max}(s_t)$ (note $A_{(i)}^{max,\pi}(s_t, a_t)$ is the same for all $i > T - t - 1$). Therefore, $\ell_n(\pi, 1) = -\mathbb{E}_{\rho^{\pi}}[\sum_{\tau=t}^{T-1} r(s_{\tau}, a_{\tau})] + f^{max}(d_0)$, which is the original RL problem. On the other hand for $\lambda = 0$, $A_{\lambda}^{max,\pi}(s_t, a_t) = A_{(0)}^{max,\pi}(s_t, a_t) = r(s_t, a_t) + \mathbb{E}[f^{max}(s_{t+1})] - f^{max}(s_t)$ in Eq (13) (we use $0^0 = 1$). We will clarify and better motivate these equations.

*L246 & L249-252.* The estimator in (14) assumes $\pi = \pi_n$ (it's a typo). We meant that $\lambda$ controls the bias and variance of (14) compared with the true gradient of (12) at $\pi = \pi_n$, as the effects of the function approximator decays as $\lambda \to 1$.

*Experiments.* The oracle policies here are obtained by prematurely terminating policy training initialized with different random seeds. We will include an ablation study of UpdateInputWhitening in the revision. We believe the result will be similar, since the only difference is in pre-training; after that, the algorithm collects the same amount of samples per iteration regardless of the number of oracle policies. We will include the range of oracle performance in the plots.

*Misc.* We can view Natural Gradient Descent as an instantiation of the first-order algorithm Mirror Descent, which uses Fisher information matrix to define Bregman divergence. AggreVaTe does not require action information, too.

## 45 Reviewer 3

*Clarity.* Thank you for the suggestions on improving the clarity! The version for general $f$ is stated in the appendix as Proposition 2 and 3. We will draw a clearer connection in the revision.

*Line 207-208.* It's a typo. We meant $\mathbb{E}_{s \sim d_0}[\max_{k \in [K]} V^k(s)] + \Delta_N - o(1)$, where $o(1)$ is due to no-regret assumption.

*Algorithm.* We sample one trajectory using the learner's policy to estimate the on-policy gradient in Eq (14). We sample the other one to learn the value function of the oracle policies, which requires the RIRO setting.

[Meta-Review · NeurIPS 2020]

The three reviewers all recommended accepting this paper. We encourage the authors to make the changes suggested by the reviewers.